



# Swell Impacts on an Offshore Wind Farm in Stable Boundary Layer: Wake Flow and Energy Budget Analysis

Xu Ning[1] and Mostafa Bakhoday-Paskyabi[1]

[1]Geophysical Institute, University of Bergen and Bergen Offshore Wind Centre

**Correspondence:** Xu Ning (xu.ning@uib.no)

**Abstract.** A parameterization of wave-induced stress is employed in an open-source large-eddy simulation code to investigate the swell impacts on the wake flow and the power output of a real offshore wind farm under a stable atmospheric boundary layer for the first time. The output module of the code is extended to include all source and sink terms of the kinetic energy equation. Two typical scenarios in the North Sea area with modest wind speeds and wind-following/opposing fast waves are considered.

Results show that swells significantly affect the profiles of wind speed and turbulence intensity across the entire operational height of the wind turbines. Such influences are prominently observed in the inflow and progressively diminish in the wake flow downstream. Through kinetic energy budget analysis, we discover that the wave effects are primarily exerted through indirect modification of the advection of energy in streamwise and vertical dimensions instead of the direct wave-induced energy input/output. The wind shift and yawing adjustment caused by waves play a crucial role in the energy harvesting

rate, depending on the specific inflow direction and wind farm layout. The absolute wave-induced changes in wind speed and turbulence intensity progressively decrease downstream, and the relative changes in total power production reach up to $20.0\%/-27.3\%$ for the wind-following/opposing wave scenarios respectively.

## 1 Introduction

As a substantial and environmentally friendly energy resource, offshore wind energy offers a promising opportunity to mitigate

climate change and accelerate the global energy transition from fossil fuels toward sustainable energy sources. Offshore wind exhibits different characteristics from its counterpart over land due to the ubiquitous and changeable wind-wave interactions at the air-sea interface, which significantly influences the airflow above through complex physical processes such as velocity and pressure perturbations, wave breaking and spray, etc (Sullivan and McWilliams, 2010). As the offshore wind industry is expected to experience rapid growth in the near future, it is crucial to enhance our understanding of Marine Atmospheric

Boundary Layer (MABL) characteristics and their impacts on the performance of offshore wind farms.

Numerous observational studies have provided evidence supporting the significant dependence of atmosphere-ocean coupling on wind-sea conditions (Donelan et al., 1997; Drennan et al., 2003; Smedman et al., 2003). The wind-sea condition is usually quantified by wave age, defined as the ratio of wave phase speed to the wind speed at 10 m height (or friction velocity), and categorized into two regimes: wind wave (young sea) and swell (old sea). Wind waves, generated locally by wind,

normally align with the wind direction and exhibit relatively small wavelengths and wave heights, depending on the wind's



duration and fetch. Swells, on the other hand, are waves originating from distant weather systems and have traveled away from their source. These waves are characterized by longer wavelengths, greater amplitudes, and higher propagation speeds than wind-sea waves. Therefore, Swell waves typically possess higher energy and are associated with more complex physical processes. These processes include the upward transfer of momentum flux (Grachev and Fairall, 2001; Kahma et al., 2016), the occurrence of low-level jets (Hanley and Belcher, 2008; Semedo et al., 2009), and surface stresses misaligned with the wind (Zou et al., 2019; Patton et al., 2019; Chen et al., 2020a). The impact of swell on the wind field is particularly pronounced under stable atmospheric conditions, where buoyancy suppresses turbulence, allowing wave-induced flow structures to dominate near the ocean surface (Zou et al., 2018; Jiang, 2020). Furthermore, a statistical analysis based on data from the 45-year European Centre for Medium-Range Weather Forecasts Re-Analysis (ERA-40) revealed that swell waves prevail across the global ocean (Semedo et al., 2011).

The Computational Fluid Dynamics (CFD) technique plays a crucial role in analyzing wind farm performance in offshore environments, primarily due to its unparalleled ability to provide detailed three-dimensional flow data. This level of detail is crucial for comprehending the complex dynamics of wind farm airflow, facilitating advancements in wind energy engineering models, including the development of more accurate reduced-order and surrogate models (Breton et al., 2017). However, most previous LES studies on offshore wind farms have applied the Monin-Obukhov Similarity Theory (MOST) with a constant roughness length near the ocean surface (Wu and Porté-Agel, 2015; Dörenkämper et al., 2015; Sood et al., 2022), an approach that may be ineffective in old sea conditions (Liu et al., 2022). Furthermore, this method was intrinsically not able to effectively reflect swell-induced upward momentum fluxes (Wu and Qiao, 2022; Ning et al., 2023). The shortcomings of roughness length parameterization can be addressed by using wave-phase-resolved LES. This approach utilizes a coordinate transformation technique to explicitly incorporate the dynamic effects of surface elevation. Yang et al. (2014, 2022b) employed this type of solver to investigate the swell impacts on the wake flows and energy harvesting. Their findings revealed that swell waves can substantially affect wake recovery and power production, by modifying both the wind profile and Turbulence Intensity (TI). Nevertheless, the usage of this wave-phase-resolved approach has been limited to simulating small wind turbine arrays in a neutral ABL, due to its high computational demands and complexity.

With the expansion of offshore wind farms and the emergence of wind farm clusters, new flow phenomena and character-istics around the wind farms have been observed and investigated. These include, but are not limited to, the blockage effect (Sanchez Gomez et al., 2023), the influence of gravity waves (Allaerts and Meyers, 2018), changes in atmospheric pressure gradients (Antonini and Caldeira, 2021) and wind farm wake deflection (van der Laan and Sørensen, 2017). The wave impact on the wind farms was studied in meso-scale in the work of Porchetta et al. (2021) and Bakhoday-Paskyabi et al. (2022). The former simulated 1250 offshore wind turbines using both the stand-alone atmospheric model (WRF) and the atmosphere-ocean coupled model (WRF-SWAN) and demonstrated a 20% difference in power output and a 25% change in wake length for cases with and without considering dynamic wave effects. The latter compared the online wind-wave coupling via WRF-SWAN and offline stand-alone coupling by WRF and showed a better performance in offline coupling with less computational cost, particularly for high wind periods. Despite this, there has been limited micro-scale research dedicated to understanding the role of waves, especially swells, in the dynamics of large-scale offshore wind farm flow problems.



In this study, we integrate a novel wave-induced stress parameterization method into a Large-Eddy Simulation code to simulate a large-scale offshore wind farm under a stable atmospheric boundary layer with low wind speed and fast swell waves. Our objective is to investigate the impacts of swells on the performance of the wind farm and its wake flow, particularly aiming to uncover the underlying mechanisms by closely examining the conservation of kinetic energy. To facilitate this, we have enhanced the PALM output module to encompass all the kinetic energy equation's source and sink terms, enabling a comprehensive budget term analysis. The paper is organized as follows: Section 2 describes the modeling tool, wave parameterization method, and detailed configuration. Section 3 presents the modeling results and their analysis. The findings of this study are summarized and concluded in Section 4.

## 2 Methods

### 2.1 LES model

In our study, we utilize the Parallelized Large-Eddy Simulation Model (PALM), an open-source LES code developed by the PALM group at Leibniz University Hannover (Maronga et al., 2020). This code is written in Fortran language and specifically designed for massively parallel computing tasks. It solves the incompressible Navier-Stokes equations as follows:

$$\frac{\partial u_j}{\partial x_j} = 0\,, \tag{1}$$

$$\frac{\partial u_i}{\partial t} = -\frac{\partial u_i u_j}{\partial x_j} - \epsilon_{ijk} f_j u_k + \epsilon_{i3k} f_3 u_{g,k} - \frac{1}{\rho_a}\frac{\partial \pi^*}{\partial x_i} + g\frac{\theta - \langle\theta\rangle}{\langle\theta\rangle}\delta_{i3} + \frac{\partial \tau_{t,ij}}{\partial x_j} + \frac{\partial \tau_{w,ij}}{\partial x_j} + S_i\,. \tag{2}$$

Here, $t$ denotes time; $u_i$, $u_j$, $u_k$ are the velocity components; $\pi^*$ is the modified perturbation pressure; $\theta$ represents potential temperature with horizontal averaging indicated by angular brackets. The subgrid-scale (SGS) turbulent stress $\tau_{t,ij}$ is parameterized by the 1.5-order Deardorff subgrid-scale model (Deardorff, 1980). The Coriolis parameter $f = (0, 2\Omega\cos\phi, 2\Omega\sin\phi)$ involves Earth's angular velocity $\Omega = 0.729 \times 10^{-4}\,\mathrm{rad\,s^{-1}}$ and latitude $\phi$ set at $54°$ (corresponding to the position of the studied wind farm). Gravitational acceleration is $g = 9.81\,\mathrm{m\,s^{-2}}$ and $\rho_a$ represents dry air density. $\epsilon$ and $\delta$ denote the Levi-Civita symbol and Dirac delta function, respectively. $S_i$ represents the momentum sink term by wind turbines. These terms align with the original governing equations in Maronga et al. (2015). However, we introduce an additional term $\tau_{w,ij}$ to denote wave-induced stress, detailed in the subsequent section. Time advancement uses the third-order Runge-Kutta method, and spatial discretization employs a staggered grid with the fifth-order Wicker-Skamarock scheme for advection.

### 2.2 Parameterization of wave-induced stress

The wave-induced stress at the ocean surface, $\boldsymbol{\tau}_w(0)$, is derived by dividing the energy transfer rate between the wind and wave fields by the wave speed, $c = \omega/k$, and integrating this value over the wave spectrum (Hanley and Belcher, 2008):





$$\boldsymbol{\tau}_w(0) = \rho_w g \int\limits_0^{2\pi} \int\limits_0^{\omega_c} \frac{\boldsymbol{k}}{w} \beta E(\omega, \Theta_w) d\omega d\Theta_w, \tag{3}$$

where $\rho_w$ represents the density of water, $\Theta_w$ is the wave propagation direction, $\omega$ denotes the wave angular frequency, and $\boldsymbol{k}$ is the wave number vector. $E(\omega, \Theta_w)$ represents the wave spectrum function, and $\beta$, the wave damping rate, is calculated using the formulas from Ardhuin et al. (2010):

$$\beta = \begin{cases} \dfrac{\rho_a}{\rho_w}(2k\sqrt{2\nu\omega}) & Re < Re_c \\[2mm] \dfrac{\rho_a}{\rho_w}(16f_e\omega^2 u_{orb}/g) & Re \geq Re_c \end{cases}. \tag{4}$$

Here $f_e$ is a constant coefficient set to 0.008. The boundary Reynolds number, $Re$, is given by $Re = 2u_{orb}H_s/\nu$, where $\nu$ is the air's kinetic viscosity, $u_{orb}$ is the surface orbital velocity, and $H_s$ is the significant wave height. The critical Reynolds number, $Re_c$, used to differentiate between the viscous and turbulent states of flow near the surface, is defined as $Re_c = 2.0 \times 10^5/H_s$. The wave spectrum in the present work is defined as the empirical wave spectrum proposed by Donelan et al. (1985) multiplied by an exponential factor:

$$S(\omega) = \frac{\alpha g^2}{\omega^4 \omega_p} \exp\left[-\left(\frac{\omega_p}{\omega}\right)^4\right]\gamma^r \exp\left[\left(\frac{\omega}{\omega_0}\right)^3\right], \ r = \exp\left[-\frac{(\omega-\omega_p)^2}{2\sigma^2\omega_p^2}\right], \tag{5}$$

where

$$\alpha = 0.006\left(\frac{U_{10}}{c_p}\right)^{0.55}, \ \gamma = 1.7 + 6.0\log\left(\frac{U_{10}}{c_p}\right), \ \sigma = 0.08\left[1.0 + 4.0\left(\frac{c_p}{U_{10}}\right)^3\right]. \tag{6}$$

Here, $\omega_p$ is the peak wave frequency and $U_{10}$ denotes the wind speed at 10 m height. $\omega_0^{-3}$ within the exponential factor is set to $-0.01$ following Hanley and Belcher (2008). This setting helps to approximate a swell-dominated wave spectrum, characterized by the dampening of high-frequency components due to dissipation over long distances. We employ a theoretical directional spectrum, expressed as

$$D(\Theta_w) = \frac{1}{\pi}\cos^2\left(\frac{\Theta_w - \Theta_{w,p}}{2}\right), \tag{7}$$

to characterize the directional distribution of wave energy, where $\Theta_{w,p}$ represents the peak wave propagation direction. The directional wave spectrum is then calculated by multiplying Eq. (5) with Eq. (7), resulting in $E(\omega, \Theta_w) = S(\omega)D(\Theta_w)$. We define a critical frequency as in Semedo et al. (2009), $\omega_c$, which demarcates the boundary between swell and wind wave in the frequency domain, calculated as $\omega_c = g/U_{10}$. The integration is performed within the range $0 < \omega < \omega_c$ to explicitly calculate the momentum fluxes from swell waves to the wind field, while higher frequency wave contributions, only acting as surface drag, are accounted for using the roughness length. Assuming that the wind and wave conditions are consistent across the domain, the wave-induced stress is treated as horizontally homogeneous, thereby making it a function dependent exclusively on height. Moreover, numerous researchers have observed that wave-induced stress decreases exponentially with





height (Högström et al., 2015; Wu et al., 2018). Therefore, we approximate the vertical profile of $\boldsymbol{\tau}_w$ by multiplying its surface
value by an exponential decay function:

$$\boldsymbol{\tau}_w(z) = \boldsymbol{\tau}_w(0)e^{-a\hat{k}z}, \ \hat{k} = \frac{\int_0^{2\pi}\int_0^{\omega_c} kE(\omega,\Theta_w)d\omega d\Theta_w}{\int_0^{2\pi}\int_0^{\omega_c} E(\omega,\Theta_w)d\omega d\Theta_w} \tag{8}$$

where $a = 1.0$ is the decay coefficient and $\hat{k}$ is the integration-weighted average wave number.

### 2.3   Wall-stress model

In our simulation, we utilize a wall-stress model that assumes a constant flux layer near the surface to estimate momentum
fluxes at the bottom of the model domain. This model differs from LES wall-stress models based on the conventional Monin-
Obukhov Similarity Theory (Monin and Obukhov, 1954), as it accounts for not only viscous stress $\boldsymbol{\tau}_\nu$ and turbulent stress $\boldsymbol{\tau}_t$
but also the stress arising from wind-wave interaction $\boldsymbol{\tau}_w$, i.e.

$$\boldsymbol{\tau}_{tot} = \boldsymbol{\tau}_\nu + \boldsymbol{\tau}_t + \boldsymbol{\tau}_w. \tag{9}$$

The sum of the viscous and turbulent stresses can be approximated by the viscosity model within the height of 10 m (Chen
et al., 2020b):,

$$\boldsymbol{\tau}_{tot} - \boldsymbol{\tau}_w = \rho_a K_m \frac{d\boldsymbol{u}(z)}{dz}, \ z < 10.0\,\mathrm{m}, \ K_m = \kappa z u_*, \tag{10}$$

where $K_m$ is the momentum eddy diffusivity. It is parameterized to be linearly proportional to both the height and the friction
velocity $u_*$, with $\kappa = 0.4$ representing the von Kármán constant. Integrating Eq. (10) yields the vertical velocity profile within
the constant flux layer:

$$\boldsymbol{u}(z) = \frac{\boldsymbol{\tau}_{tot}}{\rho_a \kappa u_*}\left[\ln\frac{z}{z_0} - \Psi_m\left(\frac{z}{L}\right)\right] - \int_{z_0}^{z} \frac{\boldsymbol{\tau}_w(z)}{\rho_a \kappa z u_*}dz. \tag{11}$$

Here, the roughness length is determined using Charnock's method (Charnock, 1955) as $z_0 = \alpha_c u_*^2/g$, where $\alpha_c = 0.012$ is the
Charnock coefficient. The first term on the right-hand side of Eq. (11) aligns with the MOST-based logarithmic wind profile,
while the second term represents the adjustment to wind velocity due to the influence of swell waves.

### 2.4   Simulation setup

In this study, we focus on a specific wind farm within a cluster situated approximately $60\,\mathrm{km}$ north of the German coast in
the North Sea. This wind farm, located at $54°30'$ N, $6°22'$ E, as indicated in Fig. 1, consists of $80$ wind turbines, each with a
capacity of $5\,\mathrm{MW}$. The wind turbines are represented by the Actuator Disk Model with Rotation (ADM-R), as detailed in (Wu
and Porté-Agel, 2015). We utilize the design parameters of the benchmark NREL-$5\,\mathrm{MW}$ wind turbine (Jonkman et al., 2009),
which includes a rotor diameter of $D = 126.0\,\mathrm{m}$, a hub height of $90\,\mathrm{m}$, and a rated wind speed of $11.4\,\mathrm{m\,s^{-1}}$.

We first perform precursor simulations (preruns) in the absence of the wind farm to obtain stable boundary layer flows. We
are interested in the regime characterized by moderate wind speeds coupled with fast-propagating waves, a scenario where



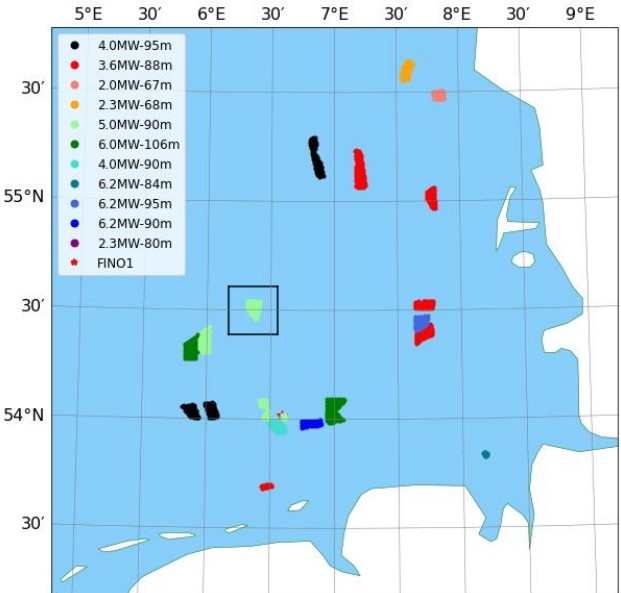

**Figure 1.** Locations of wind farms in the North Sea.

the influence of swells is relatively pronounced as reported by Chen et al. (2019); Zou et al. (2019). This choice of setup was widely used in previous numerical studies of the impact of swells on the marine atmospheric boundary layer (Sullivan et al., 2008; Nilsson et al., 2012; Jiang et al., 2016). Wind and wave data, collected from the FINO1 platform (indicated by a red star

in Fig. 1) between May 2015 and April 2016, are presented as rose diagrams in Fig. 2. The diagrams show that low to moderate wind speeds (represented by the dark blue region) primarily originate from the northwest and east, and fast peak wave speeds (indicated by the dark red region) most frequently come from the west and the direction between northwest and north. Drawing from this data, two distinct scenarios are chosen for our analysis: 1) a northwest wind with a hub-height speed of $5.0\,\mathrm{m\,s^{-1}}$, accompanied by waves originating from a $337.5°$ direction and having a peak phase speed of $12.0\,\mathrm{m\,s^{-1}}$, representing Wind-

Following Wave (WFW) condition; and 2) an easterly wind of $5.0\,\mathrm{m\,s^{-1}}$, coupled with oppositely propagating waves from west to east at a speed of $10.0\,\mathrm{m\,s^{-1}}$, as Wind-Opposing Wave (WOW) condition. These two cases are labeled as M2 and M3 respectively.

The computational domain for the preruns is set with a uniform horizontal grid size of $\Delta_x = \Delta_y = 6\,\mathrm{m}$. This grid resolution is considered sufficiently fine according to (Sanchez Gomez et al., 2023). Vertically, the grid size is $\Delta_z = 6\,\mathrm{m}$ up to a height of

$216\,\mathrm{m}$, above which it increases regularly at a rate of $1.036$, reaching the domain top at $4.6\,\mathrm{km}$ with a maximum $\Delta_z$ of $48\,\mathrm{m}$. The initial temperature profile is structured in three segments: a constant $300.0\,\mathrm{K}$ from the ocean surface to $1000\,\mathrm{m}$; a steep capping inversion up to $1200.0\,\mathrm{m}$, increasing at $1.0\,\mathrm{K}$ per $100.0\,\mathrm{m}$; and then a gradual rise by $0.1\,\mathrm{K}$ per $100.0\,\mathrm{m}$ up to the top of the boundary layer. A Rayleigh damping layer is set above $z = 1000.0\,\mathrm{m}$ to avoid the reflection of gravity waves at the upper boundary (Klemp et al., 2008). Each prerun spans $48\,\mathrm{hours}$, with the first $36\,\mathrm{hours}$ dedicated to establishing a fully developed

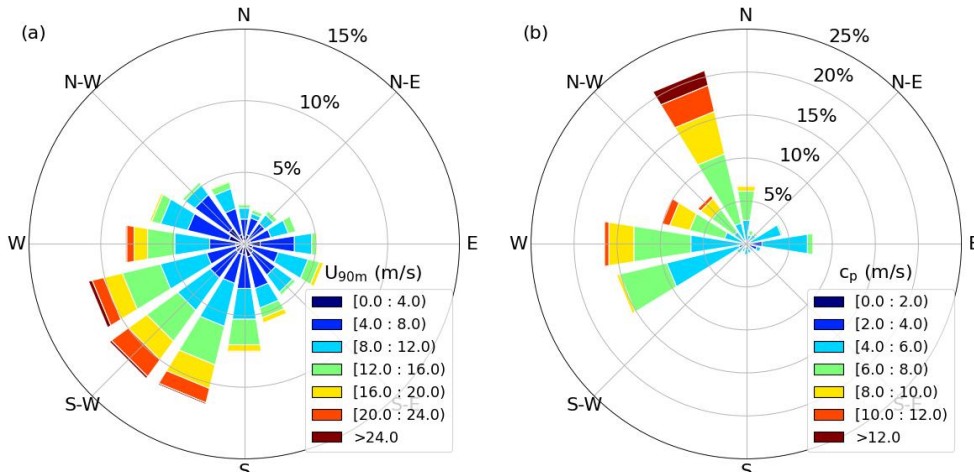

**Figure 2.** Rose diagrams of wind (a) and wave (b) in the FINO1 area, covering the period from May 2015 to April 2016.

neutral flow, followed by 12 hours of constant surface cooling at a rate of $-0.08\,\mathrm{K\,h^{-1}}$ to produce a weakly stable boundary layer. Swell-induced momentum fluxes, determined using the equations outlined in Sect. 2.2, are integrated into the wall-stress model with a roughness length of $z_0 = 0.0002\,\mathrm{m}$. This is a typical value for simulating wind waves in offshore environments. Additionally, two control cases mirroring the selected scenarios, excluding the explicitly computed wave-induced stresses, are conducted to distinctly isolate and investigate the specific impacts of the swell waves. These two control cases are labeled as M0 and M1.

In the main runs, which included the wind farm, we use the final flow data from the preruns as the initial condition, maintaining the same mesh resolution, temperature profile, and wind-wave conditions. However, the domain size is expanded to adequately simulate flows within and around the entire wind farm, as depicted in Fig. 3. To better utilize the domain, the $x$-axes are aligned with the hub-height wind direction, ensuring a consistent left-to-right wind flow in all cases. To maintain the turbulent inflow and its equilibrium with the mean wind shear and stability conditions, velocity fluctuations are continuously recycled in the region extending from $0 < x < 1.5\,\mathrm{km}$. The cyclic condition is applied to the crosswise boundaries and the outflow radiation condition is used at the outlet boundary. The wind turbines, positioned based on their real-world locations, are arranged in a right-angled trapezoidal layout within the computational domain. In cases M0 and M2, as in Fig. 3a, the wind originates from the northwest. Consequently, the $x$-axis is rotated clockwise by $45°$ from the east to align with the wind direction. The wind farm layout is adjusted accordingly: the turbine located at the northwest corner becomes the foremost in the windward direction, and the subsequent turbines are arrayed in a staggered, diagonal formation behind it, with spacings of approximately $1.1\,\mathrm{km}$ in both streamwise and crosswise directions. In cases M1 and M3 (Fig. 3b), the wind direction is easterly. Here, the turbines are arranged in columns that run west to east in line with the wind flow, with a streamwise spacing of $0.9\,\mathrm{km}$. The crosswise spacings along the south-north rows are around $0.7\,\mathrm{km}$. In both domains, a $5\,\mathrm{km}\,(40D)$ buffer is set between





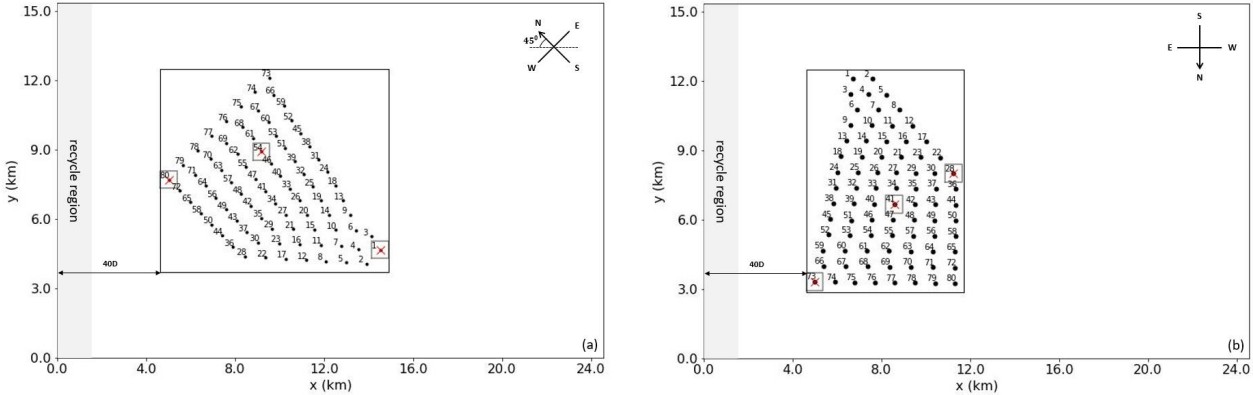

**Figure 3.** Top view of the computational domains for the main runs: wind-following wave scenario (a), wind-opposing wave scenario (b).

the inlet boundary and the wind farm to encompass the wind induction zone. Additionally, a $10\text{-}12\,\text{km}$ fetch ($80D\text{-}95D$) is allocated beyond the wind farm to the outlet boundary, ensuring sufficient space for wake flow development and recovery.

## 2.5 Kinetic energy budget

The Kinetic Energy (KE) of airflow, representing the energy due to its motion, is the direct energy source for wind turbines. Understanding kinetic energy is vital for elucidating the dynamics of the marine atmospheric boundary layer, particularly its
interactions with offshore wind farms and the ocean surface beneath. In this study, we will conduct an in-depth kinetic energy budget analysis to reveal the physical processes responsible for the generation, redistribution, and dissipation of KE for the wind field inside the offshore wind park, with a special focus on the role of swell waves in KE conservation.

The mean kinetic energy consists of the Kinetic Energy of the Mean flow (KEM) and the Turbulent Kinetic Energy (TKE):

$$\overline{E_k} = \overline{E} + \overline{e} = \frac{1}{2}\left(\overline{u}_i\overline{u}_i + \overline{u_i'u_i'}\right). \tag{12}$$

Here the prime denotes the turbulent component. The conservation equation for the mean kinetic energy can be derived by multiplying Eq. (2) with $u_i$ and taking time average:

$$\frac{\partial \overline{E_k}}{\partial t} = \underbrace{-\frac{\partial \overline{u}_k \overline{E_k}}{\partial x_k}}_{A} \underbrace{-\frac{\partial \overline{u}_i \overline{u_k'u_i'}}{\partial x_k} + \frac{\partial \overline{u_i \tau_{ki}}}{\partial x_k} - \frac{\partial \overline{u_k'e}}{\partial x_k} - \frac{1}{\rho_a}\frac{\partial \overline{u_i'\pi^{*\prime}}}{\partial x_i}}_{T}$$

$$\underbrace{-\overline{u}f_3 v_g + \overline{v}f_3 u_g}_{G} \underbrace{-\frac{1}{\rho_a}\frac{\partial \overline{u}_i \overline{\pi^*}}{\partial x_i}}_{P} + \underbrace{\frac{g}{T_0}\overline{(T-T_0)w}}_{B}$$

$$\underbrace{-\overline{\tau_{ki}\frac{\partial u_i}{\partial x_k}}}_{D} + \underbrace{\overline{u_i\frac{\partial \overline{\tau}_{w,ki}}{\partial x_k}}}_{W} + \underbrace{\overline{u_i d_i}}_{F}, \tag{13}$$





where the left-hand side is the temporal change rate of $\overline{E}_k$, and the right-hand side includes 12 terms, each with a clear physical meaning. These terms are grouped as in Maas (2023), except for the additional wave-related term:


- $\mathcal{A}$: Divergence of $\overline{E}_k$ advection

- $\mathcal{T}$: Turbulent transport of $\overline{E}_k$

- $\mathcal{G}$: Energy input by geostrophic forcing

- $\mathcal{P}$: Energy input by mean perturbation pressure gradients

- $\mathcal{B}$: Energy input by buoyancy forces


- $\mathcal{D}$: Dissipation by SGS model

- $\mathcal{W}$: Energy input by wind-wave interaction

- $\mathcal{F}$: Energy sink by wind turbines

The turbulent transport term $\mathcal{T}$ can be further divided into four parts: the transport of KEM by resolved turbulent stresses (term 1 of $\mathcal{T}$), transport of $\overline{E}_k$ by SGS stresses (term 2), the transport of TKE by resolved turbulent stresses (term 3), and the turbulent transport of TKE by perturbation pressure fluctuations (term 4).


## 3   Results

### 3.1   Inflow conditions

Figure 4 presents the vertical profiles of temporally and horizontally averaged atmospheric variables from the precursor simulations. A 12-hour surface cooling creates a positive temperature gradient from the surface up to the top of the turbine rotors,
signifying a stably stratified boundary layer. This stable stratification leads to the formation of a supergeostrophic wind jet, which spans the entire rotor height, and results in a peak wind speed of $5.2\,\mathrm{m\,s^{-1}}$ at a height of $100.0\,\mathrm{m}$ in the control case. The temperature profile exhibits only minor changes due to wave impacts, whereas the distribution of vertical velocity is significantly influenced by waves as illustrated in Fig. 4b and c. In scenario WFW, the wind-following waves accelerate the wind speed near the surface, with the jet flow occurring at a lower height than in the control case. This results in a higher wind
speed at the lower portion of the rotor and a reduced speed at the upper portion. Additionally, the wind direction in WFW shifts northward by over $10°$ below the hub height. In contrast, scenario WOW shows a slight reduction in wind speed due to opposing wave effects, with the wind direction remaining nearly identical to that of the control run.

The observed variations in wind profiles are directly attributed to the wave-induced modification of momentum fluxes near the ocean surface. As depicted in Fig. 4d and e, waves generate momentum fluxes that align with their direction of propagation.
In the WFW case, the wave field neutralizes over $80\%$ of the turbulent fluxes in the $x$-direction and introduces a negative momentum flux in the $y$-direction. This is the main cause of the increased wind speed and the northerly shift in wind direction.





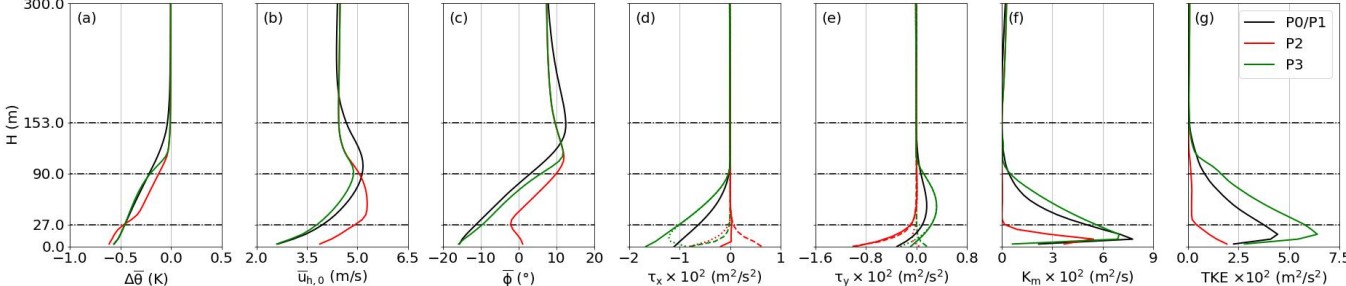

**Figure 4.** Vertical profiles of temporally (1 hour) and horizontally averaged potential temperature (a), wind speed (b), wind direction (c), momentum fluxes (dotted line, dashed line and solid line represent turbulent flux, wave-induced flux, and total flux respectively) (d,e), momentum eddy diffusivity (f), and turbulent kinetic energy (g) from the precursor simulations. The black line represents the control case without wave effects, and the red and green lines denote the cases with wind-following and wind-opposing waves respectively. The dotted-dashed lines mark the wind turbine rotor's bottom, center, and top.

Similar results were also observed in phase-resolved LES by Sullivan et al. (2008). Conversely, in the WOW case, surface stresses are enhanced due to the opposing waves. These wave effects also manifest in the turbulent characteristics of the airflow. Changes in wind shear alter the parameterized momentum eddy diffusivity $K_m$, subsequently affecting turbulence quantities.

Fig. 4f and g show that in the WFW scenario, $K_m$ is reduced to nearly zero at the rotor's bottom height, and the turbulence almost disappears beyond this height. In contrast, the WOW scenario shows a significant increase in TKE throughout the boundary layer.

To sum up, the presence of swells directly influences the momentum exchange at the wind-wave interface, and consequently, the magnitude and direction of surface stresses. This leads to a remarkable modification of the wind shear and veer close to the

waves. Although wave-induced momentum fluxes reduce significantly, by approximately $96\%$ at a height of half the wave's wavelength as by Eq. (8), these near-surface variations are further extended upwards beyond the operational height of the wind farm. This upward spread occurs through turbulent mixing as a new equilibrium is established within the entire Ekman layer, highlighting the profound influence of swells on wind farm aerodynamics.

### 3.2 Wind farm

#### 3.2.1 Flow field

Figure 5 illustrates the 1h-averaged flow field quantities including wind speed, wind direction, and turbulence intensity for case M2 (left column) and its differences from the control case M0 (right column) at the hub height horizontal plane. The turbulence intensity is defined as

$$\text{TI} = \frac{\sqrt{\frac{1}{3}(\overline{u'u'} + \overline{v'v'} + \overline{w'w'})}}{\overline{u}_{h,0}}, \tag{14}$$



**Figure 5.** Mean wind speed (a), mean wind direction (c), and turbulence intensity (e) at hub height horizontal plane for case M2. Subplots in the right column (b, d, f) are differences of corresponding quantities between case M2 and its control case M0. The solid black lines with arrows are streamlines.

where $\overline{u}_{h,0}$ is the inflow velocity. The wake region behind the wind farm is distinctly characterized by the reduced wind speed, change of wind direction, and significantly enhanced turbulence intensity. As observed in Fig. 5c and through streamlines, the wind within the wake zone undergoes a gradual counterclockwise rotation, leading to a directional shift of approximately $-10°$ at the domain's outlet. This phenomenon of wake deflection is attributed to the decrease in Coriolis force (which is proportional to wind speed) in the wake and is typically observed in large-scale wind farms, as noted by Maas and Raasch (2022).

In comparison to case M0, introducing waves in case M2 results in a slight decrease in inflow wind speed at hub height. Immediately downstream of each wind turbine, there is an acceleration of wind speed, while the waves do not significantly impact the wind speed in the farther wake region. A notable effect caused by wind-following waves is the clockwise rotation (positive $\Delta\overline{\phi}$) of the wind in both the inflow and the wake flow. As explained in Sect. 3.1, this is because the wave-induced



**Figure 6.** Same as in Fig. 5 but for case M3 and the differences from case M1.

stress alters the original Ekman equilibrium among the pressure gradient, turbulent stress, and the Coriolis force, and the wind
direction has to shift to reach a new balance. Furthermore, the turbulence intensity of the inflow shows a reduction relative
to the case without wave influence, as wave-induced stress partially offsets the surface friction. However, the TI in the wake
region remains largely unchanged, regardless of the presence or absence of waves.

Figure 6 presents the same flow field information as Fig. 5 but for M3 and its differences from M1. In M3, the presence
of wind-opposing waves leads to a reduction in inflow wind speed and a slight clockwise directional shift, consistent with
observations in Fig. 4b and c. While the near wake flow speed in case M3 is marginally lower than in M1, the far wake region
appears relatively unaffected by waves. However, the shift in wind direction is pronounced: the wake flow exhibits a notable
counterclockwise rotation (negative $\Delta\overline{\phi}$), as marked by the blue region in Fig. 6d. Despite the wind-opposing waves enhancing
momentum exchange and turbulence in the inflow, the turbulence intensity at hub height within both the near and far wake
regions remains at the same level as that in the control case.

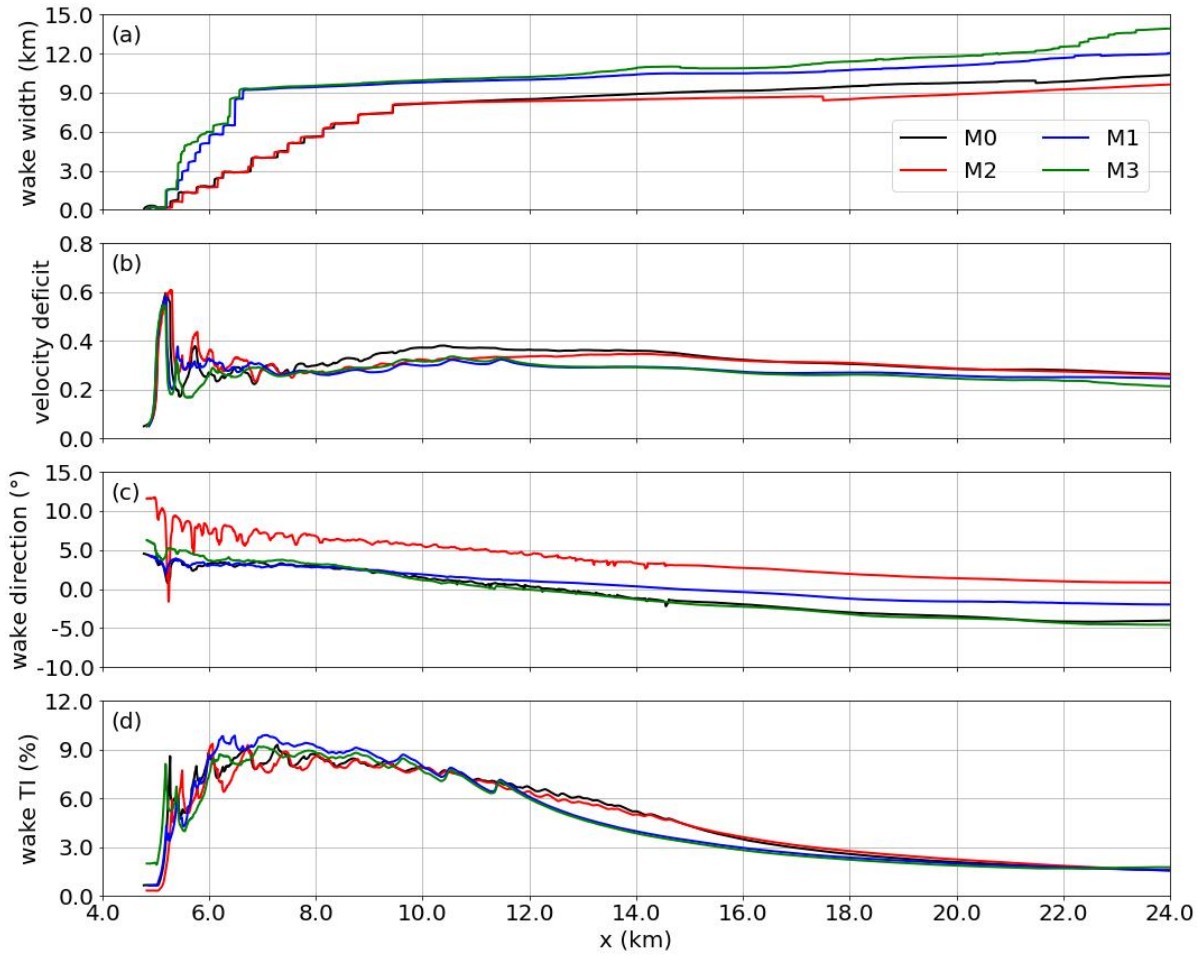

**Figure 7.** Evolution of the wake statistics along $x$-axis: wake width (a), velocity deficit (b), wake direction (c), wake turbulence intensity (d). Cases M0, M1, M2, and M3 are represented by the black solid line, blue solid line, black dashed line, and blue dashed line respectively.

To analyze the wake flow and highlight the differences between cases with and without wave effects, we defined the wake region as areas where the velocity deficit exceeds $0.05$. The velocity deficit is calculated as the relative reduction in velocity from the inflow, expressed as $1.0 - u_h/u_{h,0}$, where $u_h$ is the velocity at the hub height within the wake and $u_{h,0}$ is the inflow velocity at the same height. Downstream along the $x$-axis, we computed the wake width by measuring the span between its left and right wake edges. Additionally, we evaluated the velocity deficit, wind direction, and turbulence intensity, averaged across

the wake width at each $x$-position. This enables a detailed examination of how waves impact wake development. These wake statistics along the $x$-axis for all cases are illustrated in Fig. 7.

    Figure 7a indicates that within the wind farm, the wake width remains unaffected by wave conditions. However, further downstream, the wake width is influenced depending on the wave direction: it is narrowed by wind-following waves and



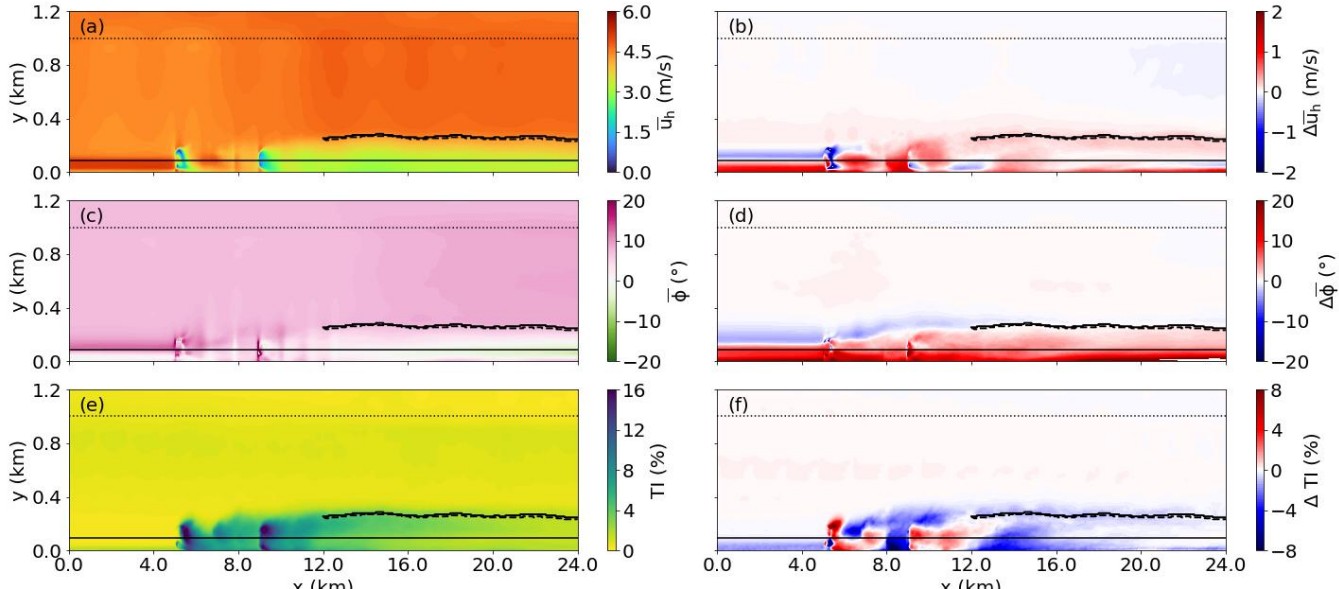

**Figure 8.** Mean wind speed (a), mean wind direction (c), and turbulence intensity (e) at the central $x$-$z$ plane for case M2. Subplots in the right column (b, d, f) are differences of corresponding quantities between case M2 and its control case M0. The solid black line and the dotted line are the hub height and the bottom of the inversion layer. The dashed line marks the top of the internal boundary layer.

broadened by wind-opposing waves. This variation in wake width due to wave influence extends from a few hundred meters

just behind the wind farm to approximately $2.0\,\mathrm{km}$ near the domain's outlet. Regarding velocity deficit, simulations with and without wave effects show almost consistent results for different wave directions, with only minor fluctuations observed within the wind farm. In contrast, the impact of waves on wind direction is more significant. In case M2, the wake direction shifts by over $5°$ compared to M0, while in M3, the shift is about $-3°$ relative to M1. A directional shift of $3° \sim 5°$ implies a crosswise wake deviation of approximately $50.0$ to $120.0$ meters at a normal streamwise spacing in a wind farm, which could lead to a

strong impact on the total wind farm power output. Furthermore, the turbulence intensity at hub height appears to be minimally influenced by wave conditions in both scenarios. Across all cases, TI exhibits a consistent trend in the streamwise direction: it increases sharply at the front part of the wind farm, slightly decreases until the last row of wind turbines, and finally, slowly reverts to the ambient level in the distant wake region, typically beyond the extent of one wind farm length.

    Figures 8 and 9 present the time-averaged flow statistics on the $x - z$ plane through the wind farm's center. In both M2 and

M3 cases, an Internal Boundary Layer (IBL) begins to form immediately behind the first row of turbines. In contrast to the growth pattern seen in Conventionally Neutral Boundary Layers (CNBL), where the IBL's vertical extent can reach heights of $3D$ to $4D$ downwind (Allaerts and Meyers, 2017), our scenarios show a different behavior. In our simulations, the cold air at the lower part of the boundary layer is entrained and drawn upward under the mixing effect of the wake turbulence, forming





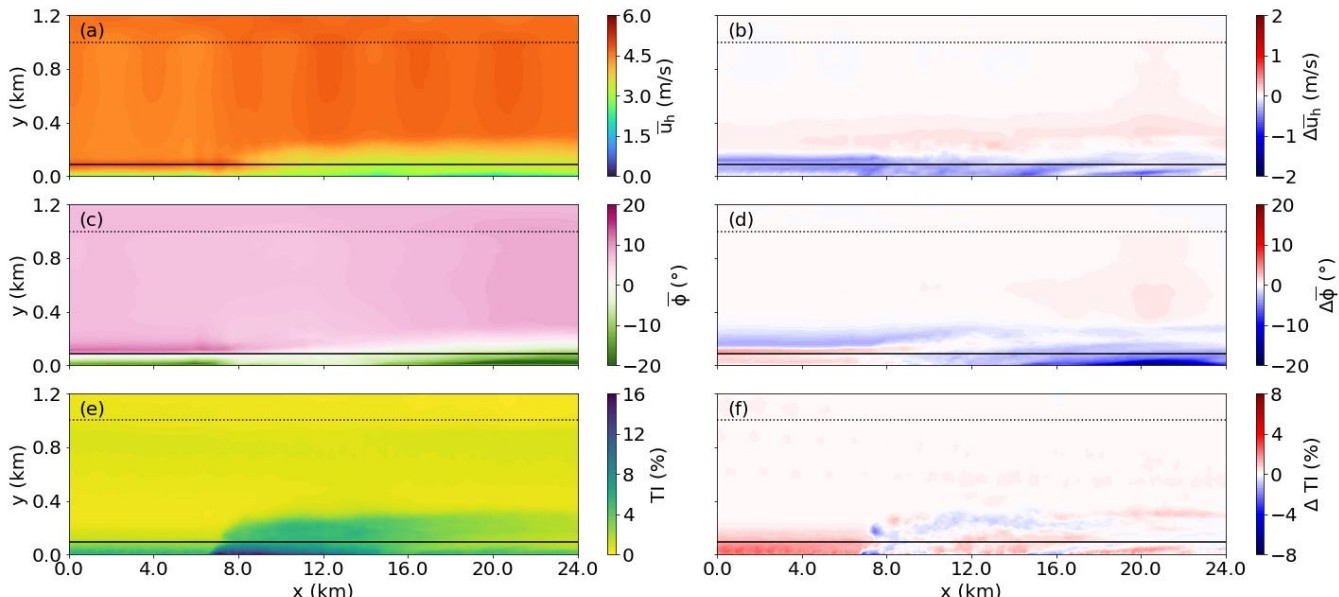

**Figure 9.** Same as in Fig. 8 but for case M3 and the differences from case M1.

a sharper temperature gradient at the top of IBL. This in turn restrains further expansion of IBL, which ceases its growth at
approximately $2D$ height, indicated by the black dashed lines.

The heights of IBL in the control cases along $x$-direction align closely with those in the wave-affected scenarios, indicating
that the evolution of the upper part of wake flow is hardly affected by wave conditions. However, the effects of waves on the
flow below the hub height are notably distinctive. In Fig. 8b and d, it is observed that, in case M2, the wind speed near the
surface can exceed that of M0 by $1.0$ to $1.5\,\mathrm{m\,s^{-1}}$. Additionally, a significant clockwise rotation in wind direction is evident in
the dark red region below the hub height line. Wave effects are also apparent in the comparison between cases M1 and M3, as
shown in Fig. 9. Here, under the influence of wind-opposing waves, there is a noticeable decrease in velocity below hub height,
accompanied by a counterclockwise shift in wind direction. In contrast to the enhanced TI and faster recovery of the wake of a
single wind turbine under wind-opposing wave condition (Yang et al., 2022b), the impact of waves on the turbulence intensity
within the wind farm wake flow in our study is less pronounced. This is mainly because the wake's turbulence is predominantly
mechanical turbulence originating from the wind turbines themselves, which overwhelms the wave-coherent turbulence.

### 3.2.2 Energy budget analysis

The integration of each term in the energy budget was computed over the control volume of the wind farm, denoted as $\Omega_{wf}$.
This volume extends horizontally to $3D$ beyond the wind farm's edges, as outlined by the black squares in Fig. 3. Vertically,
$\Omega_{wf}$ spans from $15\,\mathrm{m}$ to $201\,\mathrm{m}$ above the surface. The results are presented in Fig. 10. Theoretically, in an equilibrium state, the
mean kinetic energy of a flow field should remain constant, implying that the sum of the terms on the right-hand side of Eq. (13)



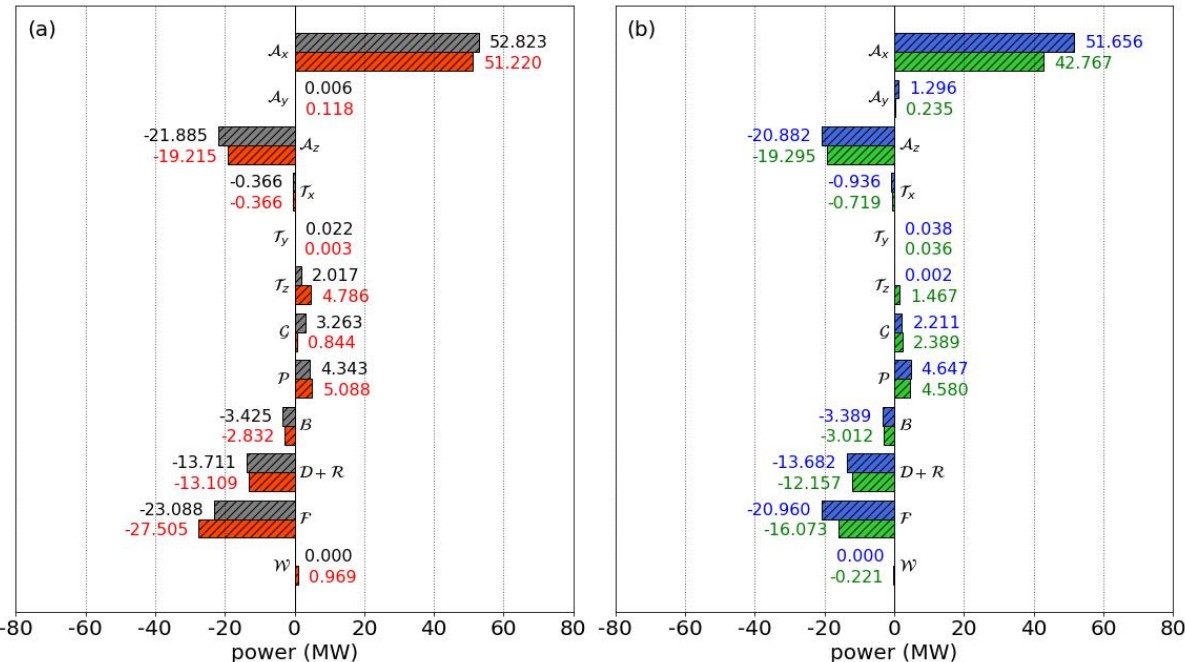

**Figure 10.** The mean kinetic energy budget terms in the wind farm control volume for M0 and M2 (a), M1 and M3 (b). M0, M2, M1, and M3 are colored grey, red, blue, and green.

would be zero. However, in our simulations, this sum yields a positive value. This discrepancy can be attributed to two primary reasons: firstly, the flow never reaches a perfectly steady state due to the continuous evolution of the potential temperature profile (as shown in Fig. 4a) driven by the imposed surface cooling setup (Sanchez Gomez et al., 2023). Secondly, there is an inherent underestimation of dissipation resulting from the numerical integration method and the computational schemes
used in the PALM code, as discussed in Maas (2022). To account for this, we combine this residual with the dissipation term, treating them as a single energy sink term, denoted as $\mathcal{D} + \mathcal{R}$.

    We find that the inclusion of swell does not qualitatively alter the energy source and sink terms. In all four cases in this study, six common terms contribute to the total energy input: $\mathcal{A}_x$, $\mathcal{A}_y$, $\mathcal{T}_y$, $\mathcal{T}_z$, $\mathcal{G}$, and $\mathcal{P}$. Among them, the advection of kinetic energy in the $x$-direction $\mathcal{A}_x$ is the predominant source of energy. Notably, the kinetic energy transports in the $y$-direction by
both the mean flow ($\mathcal{A}_y$) and turbulence ($\mathcal{T}_y$) are significantly less, typically two to three orders of magnitude smaller than $\mathcal{A}_x$. Compared to the findings in Maas (2023), where the vertical turbulent transport of $\overline{E}_k$, i.e. $\mathcal{T}_z$, is comparable to $\mathcal{A}_x$, our simulations under stable atmospheric conditions exhibit different behavior. In our cases, the turbulence is suppressed by the buoyancy force, and the wake turbulence intensity rapidly reverts to the ambient level. As a result, the vertical turbulent transport term, $\mathcal{T}_z$, is an order of magnitude smaller than $\mathcal{A}_x$, leading to a slow recovery of velocity deficit (as also shown in
Fig. 7). The common energy sink in all cases includes the vertical transport by mean flow $\mathcal{A}_z$, turbulent transport along $x$-axis $\mathcal{T}_x$ (which is negligible), buoyancy term $\mathcal{B}$, dissipation $\mathcal{D} + \mathcal{R}$, and energy extraction by the wind farm $\mathcal{F}$. It is worth noting





that while the buoyancy term $\mathcal{B}$ is relatively small in magnitude in all simulations, it can be a significant factor in strongly stable or convective stability conditions.

Compared to case M0, the magnitudes of $\mathcal{A}_x$ and $\mathcal{A}_z$ in case M2 exhibit slight reductions of $3.0\%$ and $12.2\%$, respectively. This is primarily attributed to changes in the mean wind speed profile influenced by wind-following swell effects. Additionally, despite a notable reduction in turbulence at the lower boundary of $\Omega_{wf}$ in M2, there is a substantial increase of $137.3\%$ in $\mathcal{T}_z$. This increase is caused by the combination of an upward turbulent momentum flux (negative turbulent stress) and a negative wind shear induced by waves. A sharp decrease in the geostrophic term $\mathcal{G}$ is mainly due to the clockwise shift of the wind direction, resulting in a negative $\overline{v}$. Notably, the contribution to $\overline{E}_k$ from the wave-induced stresses itself is relatively minor, only constituting about $1.5\%$ of the total energy input. However, it leads to a $19.1\%$ increase in energy extraction by the wind farm. This implies that the impact of waves on the wind farm's energy budget is primarily indirect, through modifications to the mean wind speed and direction, rather than from the wave-induced stresses themselves. Variations in other terms are of a magnitude of $0.1\,\mathrm{MW}$ for 80 wind turbines in total and thus are considered insensitive to the presence of swells.

In case M3, the presence of wind-opposing waves results in a reduction of wind speed at various levels throughout the rotor range, leading to a decrease of $17.2\%$ in $\mathcal{A}_x$ and $7.6\%$ in $\mathcal{A}_z$ compared to case M1. $\mathcal{T}_z$ shows a remarkable increase, but unlike in case M2, this increase is due to the enhanced kinetic energy entrainment across the upper $\Omega_{wf}$ boundary. The geostrophic term $\mathcal{G}$ remains largely unchanged, as the wind direction is not significantly affected by the opposing waves. The energy sink related to $\tau_w$ is minimal, at only $-0.2\,\mathrm{MW}$ ($0.4\%$ of the total sink). However, the indirect effects of the waves lead to a substantial $23.3\%$ reduction in energy extraction by the wind farm.

Figure 11 shows the power density profiles of budget terms (except for those in $y$-dimension). The integration of these profiles along the $z$-axis gives the corresponding budget values as in Fig. 10. It provides a clear view of the wave effects on the $\overline{E}_k$ budget terms at various height levels. Figure 11a illustrates highly consistent profile shapes of $\mathcal{A}_x$ and wind speed (see also Fig. 4b), indicating again that $\mathcal{A}_x$ is largely determined by the inflow wind profile. Due to the presence of a velocity deficit in the wake, the inflow at the $x$-axis boundaries of $\Omega_{wf}$ exceeds the outflow, causing the mean flow to diverge through the top and bottom planes. Consequently, any acceleration of wind speed at these boundaries results in a greater amount of kinetic energy being carried away, and vice versa. This is the reason for the larger magnitude of $\mathcal{A}_z$ near the surface and a smaller one at the rotor top in M2 compared to M0. The wind-following wave condition in M2 induces negative wind shear and alters the direction of turbulent momentum flux, accounting for the increased $\mathcal{T}_z$ across the rotor, while the larger $\mathcal{T}_z$ in M3 compared to M1 is mainly observed at the upper part of the rotor. Figure 11d demonstrates the influence of wind direction on the geostrophic term $\mathcal{G}$. A clockwise shift of wind leads to a reduction in $\mathcal{G}$. As for the energy contributions from mean perturbation pressure $\mathcal{P}$, and the energy sinks due to buoyancy $\mathcal{B}$ and dissipation $\mathcal{D} + \mathcal{R}$, these terms are not sensitive to the presence of waves in both cases M2 and M3, even near the surface. Figure 11i displays the exponential decay of $\overline{E}_k$ rate of change directly caused by the wave-induced stresses, which die out quickly above the height of $z = -0.5\mathrm{D}$.



**Figure 11.** The power density vertical profiles of each mean kinetic energy budget terms for M0 (black), M2 (red), M1 (blue), and M3 (green).

## 3.3 Wind turbine

### 3.3.1 Flow field

While the previous section addressed the impact of waves on the overall wind farm, this section explores the variation of wave effects among individual wind turbines at different positions, intending to provide a clearer picture of the wave-influenced airflow dynamics inside the wind farm. To achieve this, we chose three representative turbines in each case (marked by cross signs in Fig. 3): turbines No. 80, No. 54, and No. 1 for cases M0 and M2, and turbines No. 73, No. 41, and No. 28 for cases



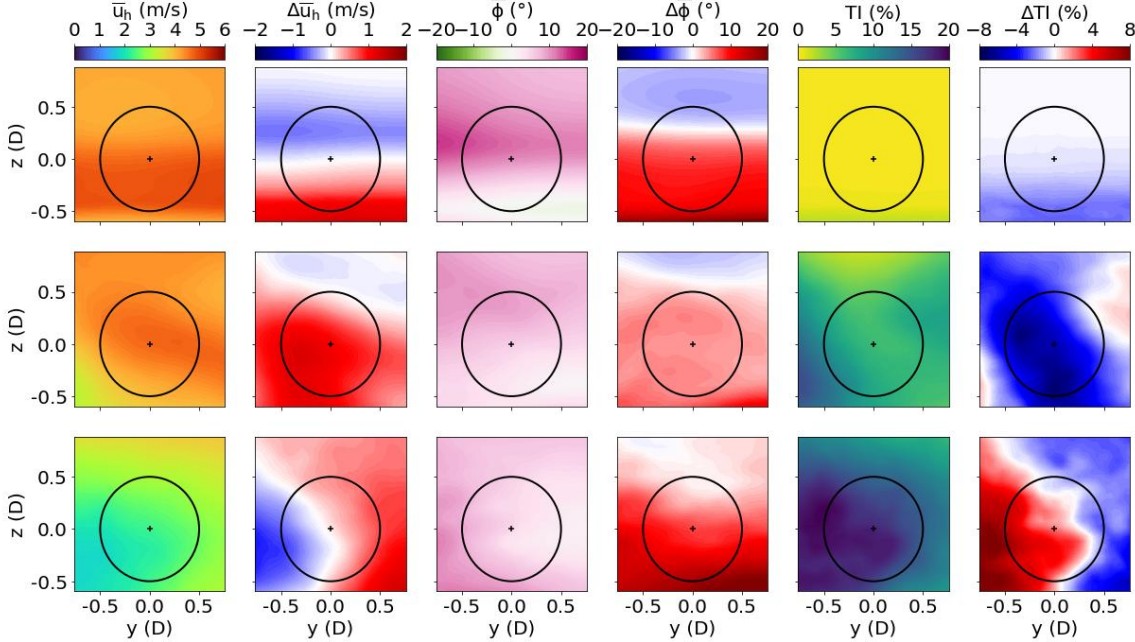

**Figure 12.** Flow statistics in the rotor planes of three wind turbines at the front, middle, and back positions in the wind farm for case M2 and the differences between M2 and M0. Columns 1, 3, and 5 show the mean horizontal wind speed, wind direction, and turbulence intensity respectively. Columns 2, 4, and 6 are the differences of the corresponding quantities from case M0.

M1 and M3. These turbines are selected to represent the front (WT-F), middle (WT-M), and back (WT-B) segments, offering a comprehensive view of the wave effects across the entire wind farm.

Figure 12 plots the mean horizontal wind speed (column 1), wind direction (column 3), and turbulence intensity (column 5) for the wind turbines at the front (row 1), middle (row 2), and back parts (row 3) of the wind farm in case M2, and the corresponding differences between cases M2 and M0 are shown in columns 2, 4, and 6. WT-F reflects the conditions experienced by the first-row turbines, which are subject to swell impacts similar to those on the ambient inflow, as shown in Fig. 4. The wind speed's acceleration and the clockwise shift in wind direction at the lower rotor section, due to wind-following waves, are evident in Fig. 12b and d, where TI is also observed to be slightly lower near the surface. WT-M benefits from this altered wind direction and remains unobstructed by upstream wake flow. Consequently, there's a notable increase in wind speed and a reduction in TI throughout the rotor area compared to the corresponding turbine in case M0. WT-B's rotor is partially covered by wake, resulting in higher wind speeds and reduced turbulence on the right half of the rotor compared to the left. Notably, the influence of waves on airflow inside the wind farm progressively extends from the lower rotor area in the front row to higher altitudes in the middle, eventually impacting the entire rotor area in the farm's rear region.

The most pronounced influence of wind-opposing waves is the intensification of turbulence. In case M3, WT-F encounters stronger turbulence across the entire rotor compared to its counterpart in M1, as seen in Fig. 13f. The wind speed is slightly





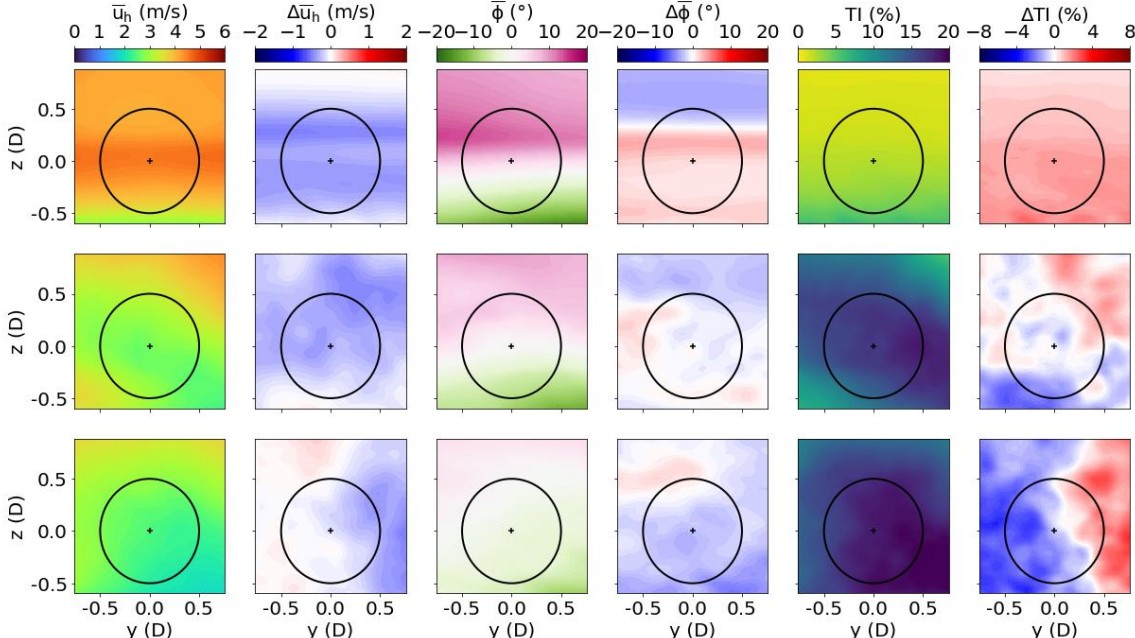

**Figure 13.** Same as in Fig. 12 but for case M3 and the differences from case M1.

reduced, while the wind direction is barely changed, contrasting with conditions under wind-following waves. The wave-induced variations in flow quantities gradually decay further downstream. The inflow direction and TI for WT-M in M3 are almost consistent with those in M1, although the decrease in velocity is still present. WT-B in the last row faces an asymmetric inflow turbulence due to its yaw adjustment to the south.

### 3.3.2 Energy budget analysis

With the same analysis method as discussed in Sect. 3.2.2, but focusing on the individual wind turbine control volume $\Omega_{wt}$ as shown by grey squares in Fig. 3, we calculate the mean kinetic energy budget terms for three selected turbines in each simulation case and demonstrate the results in Fig. 14.

As a result of the accumulating velocity deficit in wake flow, the primary energy source $\mathcal{A}_x$ for a wind turbine's control volume diminishes progressively downstream, with $\mathcal{A}_x$ of WT-B being merely about $10.0\%$ of that of WT-F. This aligns with findings from previous LES studies by Allaerts and Meyers (2017) and Maas (2023). $\mathcal{A}_y$ is mainly attributed to the asymmetry of the inflow. For front-row turbines, $\mathcal{A}_y$ arises from yaw misalignment, while for turbines further back, it is mainly influenced by rotor wake interference. For instance, WT-B in cases M0 and M2 experiences partial wake obstruction, leading to a significant disparity in velocities at its left and right sides. This results in a substantially higher $\mathcal{A}_y$ for WT-B. To compensate for the KE loss through lateral boundaries, there's a corresponding increase in $\mathcal{A}_z$, contrasting with the conditions





experienced by WT-F and WT-M, where such wake obstruction is less pronounced or absent (column 2 in Fig. 12). The influence of waves on the kinetic energy advection along the $x$-axis, $\mathcal{A}_x$, becomes increasingly pronounced for wind turbines located further downstream. In case M2, which involves wind-following waves, the changes in $\mathcal{A}_x$ compared to M0 are $-3.1\%$ for WT-F, $35.6\%$ for WT-M, and $103.3\%$ for WT-B. The corresponding differences due to wind-opposing wave conditions between M3 and M1 are $-5.1\%$, $-20.0\%$, and $-71.8\%$ respectively.

The turbulent transport of $\overline{E}_k$ is marginal in magnitude compared with the contribution from advection. However, it is worth noticing that the waves result in a remarkable increase ($83.7\%$) in $\mathcal{T}_z$ for the last-row turbine in M2, while this increase in M3 is only $11.8\%$. As for the geostrophic term $\mathcal{G}$, there is a consistent trend of increase downstream in both cases. However, $\mathcal{G}$ is largely reduced in M2 due to the presence of waves. $\mathcal{P}$ signifies the work performed by perturbation pressure across the wind turbine control volume $\Omega_{wt}$. It is elevated at both the front and rear of the wind farm. This increase is due to the larger
pressure gradients present at these boundaries. In the presence of wind-following waves, $\mathcal{P}$ increases at the front row, as these waves intensify the pressure drop in the $x$-direction across the rotor. Conversely, wind-opposing waves reduce $\mathcal{P}$ by decreasing this pressure drop. The velocity changes induced by waves have a relatively minor effect on variations in $\mathcal{P}$. Downstream, $\mathcal{P}$ exhibits non-monotonic variations, potentially linked to the small-scale gravity wave oscillations identified in the study by Maas (2023). The dissipation term $\mathcal{D} + \mathcal{R}$, as expected, follows the variation trend of turbulence intensity, exhibiting a greater
magnitude within the wind farm than at its edges. Interestingly, though the change in the absolute magnitude of the energy extraction term $\mathcal{F}$ due to waves decreases downstream, the relative change is more pronounced for WT-M compared to WT-F and WT-B.

### 3.3.3   Yaw and power extraction

In practice, wind turbines work with a control system designed to optimize power output by adjusting their operational states.
Yawing control is a critical part of this system because the wake direction is largely determined by the yaw angle and the total energy production could vary a wide range with different yawing conditions given the same wind farm layout Bastankhah and Porté-Agel (2019); Munters and Meyers (2018). The yawing control module in PALM is turned on in the present study so that each wind turbine adjusts its yaw angle according to the local wind direction until the yaw misalignment threshold of $5.0°$ is reached, and the yawing speed is set to $0.3°$ per second.

Figure 15 gives a comprehensive view of how swell waves from two opposite directions affect the yawing behavior of wind turbines at various positions within the wind farm, and Table 1 details the yaw statistics for all four cases. In case M2, the first-row turbines yaw northwards to align with the clockwise-shifted wind under the wind-following swell effects, with an average yaw angle difference of about $10°$ compared to M0. This variation rapidly phases out for turbines deeper within the wind farm, as indicated by the gradual fading of the rose color from west to east in Fig. 15a. However, turbines in the rear
section (No. 1 to No. 22) exhibit larger yaw angle differences again. These discrepancies are mainly due to the wave influences on the wind farm wake deflection. The increase in wind speed below the hub height, caused by wind-following waves, mitigates the wake's velocity deficit, thereby reducing the geostrophic force responsible for wake deflection. This phenomenon becomes more evident in the downstream turbines. The pattern where yaw differences are more pronounced at the front and back of the



**Figure 14.** The mean kinetic energy budget terms in the wind turbine control volumes for M0 and M2 (a,b,c), M1 and M3 (d,e,f). M0, M2, M1, and M3 are colored grey, red, blue, and green. The left, middle, and right columns represent the front, middle, and back wind turbine positions in the wind farm.

wind farm than in the middle is also observed between M3 and M1 (as shown in Fig. 15), though with a smaller magnitude of the average yaw difference $\Delta\gamma$. The wind-opposing waves aggravate the velocity deficit across the wind farm, causing the wake flow to veer further leftward. Consequently, turbines at the back rows rotate towards the south to minimize yaw misalignment.

The differences in energy production caused by waves are illustrated in Fig. 16 and Table 2 lists the related statistics. It is worth noting that while the first-row wind turbines in M2 gain more energy than those in M0 due to the larger inflow wind speed, the maximal power increase appears in the middle region, where the turbines are less influenced by the upwind wake



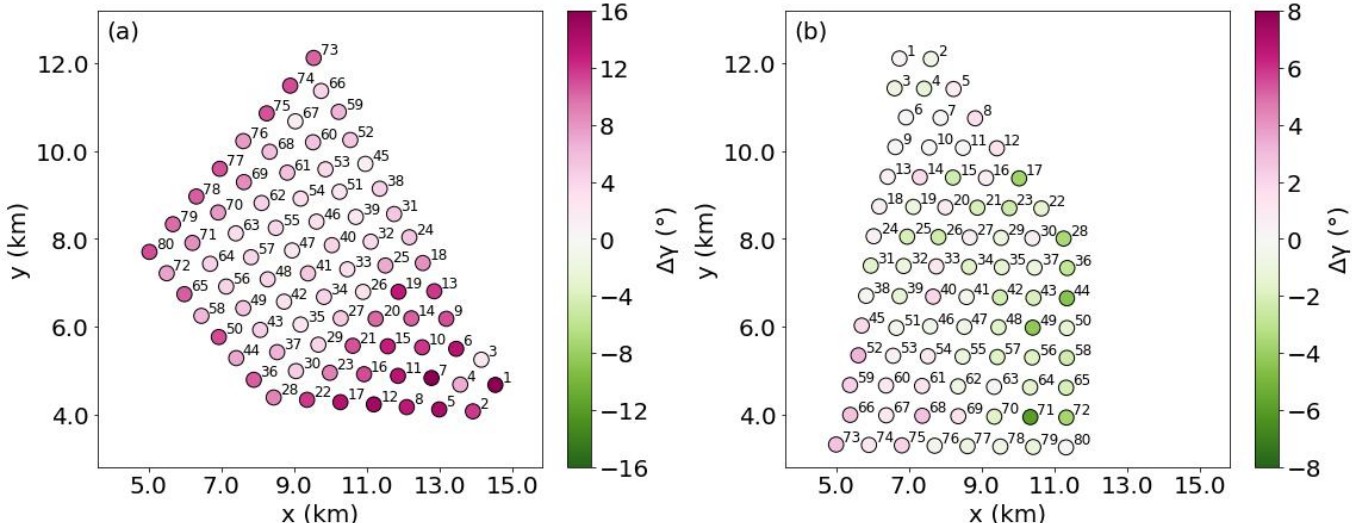

**Figure 15.** The yaw angle difference of each wind turbine between M2 and M0 (a), and between M3 and M1 (b).

**Table 1.** Wind turbine yaw angle statistics in degree. Δ signifies the difference from the corresponding control case. RMSD is the root mean square deviation.

| Case ID | Min | Max | Mean | ΔMin | ΔMax | ΔMean | RMSD |
|---------|-----|-----|------|------|------|-------|------|
| M0 | -8.2 | 5.0 | 0.6 | | | | |
| M1 | -3.7 | 5.3 | 0.5 | | | | |
| M2 | -1.2 | 15.3 | 8.1 | 0.9 | 18.2 | 7.4 | 4.0 |
| M3 | -5.2 | 5.8 | 0.0 | -5.5 | 3.2 | -0.5 | 1.8 |

flow as a result of wind shifting. The powers of turbines in the rear region are slightly increased, except for No. 3 and No. 4, which are fully covered by the shifted wake. The total energy extraction increases from $20.0$ MW to $24.0$ MW, i.e. an improvement of $20.0\%$, which is considered a substantial value for a large-scale wind farm. By contrast, when swell waves propagate against the wind, they result in a power reduction for each wind turbine, and this reduction value decreases from approximately $-0.1$ MW in the front row to $-0.01$ MW in the last row. The overall energy extraction loss in M3 due to waves

is $4.8$ MW ($-27.3\%$), with the maximal individual difference reaching $-0.13$ MW.

## 4   Discussion and conclusions

The interaction between the atmosphere and ocean waves, especially swells, has gained considerable research interest for their strong influence on the dynamics of marine atmospheric boundary layer flows and the operation of large-scale wind farms.



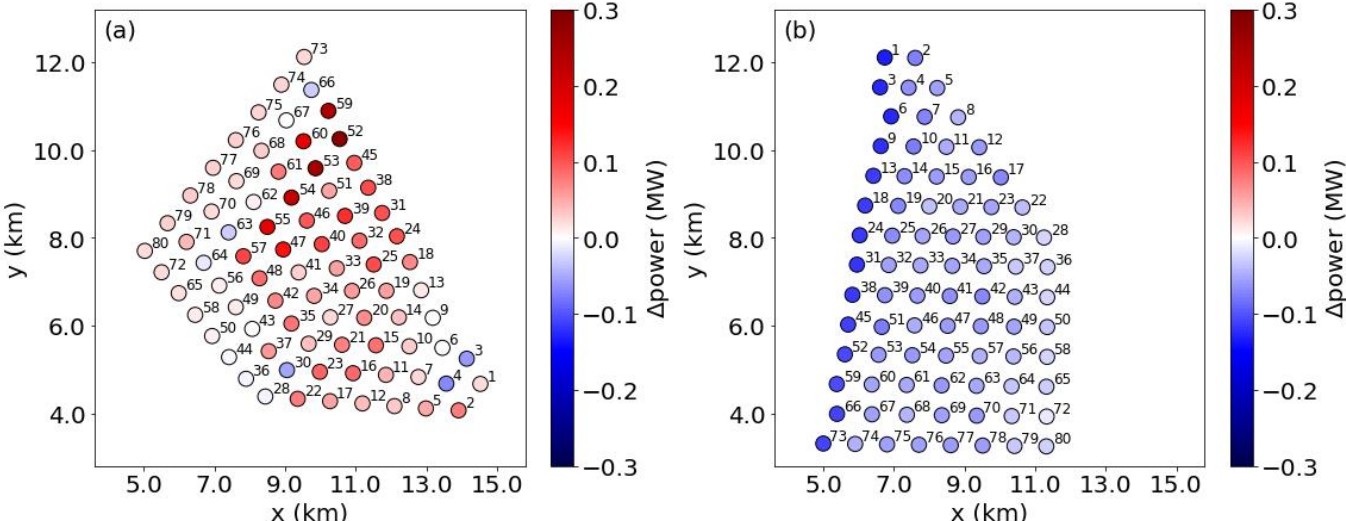

**Figure 16.** The power extraction difference of each wind turbine between M2 and M0 (a), and between M3 and M1 (b).

**Table 2.** Wind turbine power extraction statistics in MW. Δ signifies the difference from the corresponding control case. RMSD is the root mean square deviation.

| Case ID | Min | Max | Mean | ΔMin | ΔMax | ΔMean | RMSD |
|---|---|---|---|---|---|---|---|
| M0 | 0.02 | 0.55 | 0.25 | | | | |
| M1 | 0.07 | 0.53 | 0.22 | | | | |
| M2 | 0.02 | 0.58 | 0.30 | -0.07 | 0.29 | 0.05 | 0.06 |
| M3 | 0.04 | 0.41 | 0.16 | -0.13 | -0.01 | -0.06 | 0.03 |

Despite this, most prior numerical studies have been limited to neutral stability conditions and idealized wind farm layouts. To address this gap, the present large-eddy simulation study focuses on an actual wind farm situated at the North Sea and investigates the swell impacts on both wind farm performance and wake flow dynamics under stably stratified boundary layer flows.

We enhance the original wall-stress model in PALM to capture the effects of waves accurately with a new parameterization method. This method computes the vertical profiles of wave-induced stresses using a predefined wave spectrum, enabling it to simulate more complex wind-wave interaction scenarios. Specifically, it effectively represents upward momentum fluxes and cases involving misalignment between wind and wave directions. Based on the wind-wave data from May 2015 to April 2016, we identify two representative scenarios characterized by moderate wind speeds ($5.0\,\mathrm{m\,s^{-1}}$) and fast wave ($12.0\,\mathrm{m\,s^{-1}}$ and $10.0\,\mathrm{m\,s^{-1}}$) conditions: the first involves a northwesterly wind accompanied by a wind-following wave, with a slight misalignment angle of $22.5°$; the second includes an easterly wind opposed by a westerly originated wave. We are interested





in the wave impacts under stable stability conditions because in such cases the wind turbulence is suppressed and thus the wave-induced momentum plays a more important role in the boundary layer flows (Jiang, 2020).

In both selected scenarios, the presence of waves significantly influences the inflow characteristics. For the wind-following wave case, the waves induce an ageostrophic jet below the hub height, along with a clockwise wind shift and a notable reduction in turbulence intensity. In contrast, the wind-opposing wave scenario leads to reduced wind speeds and increased turbulence 450 intensity across the rotor, with minimal change in inflow direction. These effects are quantitatively consistent with the results of previous numerical studies conducted under neutral conditions (Sullivan et al., 2008; Jiang et al., 2016). A key distinction in wind-wave interaction under neutral versus stable conditions is the height of the stable boundary layer, which in our simulations is about $160.0\,\mathrm{m}$, significantly less than the typical $1.0\,\mathrm{km}$ height of a neutral boundary layer. This results in more pronounced wave-induced wind shear, wind veer, and turbulence intensity variations exactly within the operational height range of a wind 455 farm. This underscores the research significance of these scenarios in the context of wind farm operations.

Partly aligned with the simulation results from Yang et al. (2022b, a), the differences in wind speed and turbulence intensity due to waves are detected inside the wind farm. Compared to the case without waves, a weaker TI and thus slower recovery of velocity deficit is found in the WFW case. However, for the WOW case, there is neither a strong enhancement of TI in the wake flow nor a remarkable faster recovery. There are two main reasons: firstly, the wave heights used in this study ($1.36\,\mathrm{m}$ for 460 WFW and $0.96\,\mathrm{m}$ for WOW) are smaller than theirs ($3.2\,\mathrm{m}$); secondly, the wind farm in our case is much larger (80 turbines) compared to the wind turbine arrays in their study (6 turbines). As a result, the extra TI in the inflow induced by waves is rapidly overwhelmed by the mechanical turbulence generated by the wind turbines. Furthermore, the influence of waves on the wind speed and TI of the wind farm's wake flow diminishes rapidly as it moves downstream, and is barely distinctive in the far wake region (Fig. 7). Nevertheless, waves can significantly change the flow direction by inducing a crosswise velocity 465 component and modifying the Coriolis force within the wake. This wind direction shift persists until the domain's end and notably influences the aerodynamic performance of downstream wind turbines.

The analysis of kinetic energy budget terms over wind farm control volume $\Omega_{wf}$ reveals that waves influence the $\overline{E}_k$ balance mainly by increasing (WFW)/decreasing (WOW) mean wind speed and thus modifying the energy advection in $x$- and $z$-directions, i.e. $\mathcal{A}_x$ and $\mathcal{A}_z$. Besides, the vertical turbulent transport term $\mathcal{T}_z$ is also substantially affected. $\mathcal{T}_z$ in the WFW case 470 increases due to the wave-induced upward momentum fluxes and the negative wind shear, while the increase of $\mathcal{T}_z$ in the WOW case is a result of the enhanced turbulence at the top of $\Omega_{wf}$. However, the direct wave-induced energy term $\mathcal{W}$ is negligibly small, accounting for only $1.5\%$ and $0.4\%$ of the total power in both cases.

In addition, the aerodynamics of three wind turbines representative of the front, middle, and back regions of the wind farm are also analyzed, to investigate how the wave influences vary with positions in the wind farm. $\overline{E}_k$ budget terms analysis over 475 $\Omega_{wt}$ shows that though the absolute wave-induced wind decays quickly, the relative changes of energy advection for individual turbines increase as going downstream, e.g. the changes in $\mathcal{A}_x$ are $3.1\%/-5.1\%$ for WT-F, $35.6\%/-20.0\%$ for WT-M, and $103.3\%/-71.8\%$ for WT-B in cases WFW/WOW respectively.

While waves mainly impact the energy harvesting of wind turbines through changes in energy advection, the role of wind direction shift and corresponding yaw adjustments play a crucial role in this process. In the wind-following wave scenario, the

480 most substantial power changes occur in the middle section of the wind farm. This is primarily because the reduction in inflow wind caused by upstream wake effects is significantly mitigated for these turbines due to the wave presence. The alteration in energy production related to wind shift depends highly upon the ambient wind direction and the specific layout of the wind farm. Previous research has not paid enough attention to this aspect. Incorporating yawing control is therefore critical and should be a key consideration in future studies on this topic.

In brief, the main contributions and findings of the present work are summarized as follows:

1. A parameterization method for wave-induced stresses is for the first time incorporated with the wall-stress model in PALM to investigate the swell impacts on stable atmospheric boundary layers.

2. The output module of PALM is extended by adding KE-related quantities to reveal the mechanism of wave effects through budget analysis. Results demonstrate that the wave affects the wind farm flow not mainly by the direct work done by itself but by the indirect modification of the energy transport in $x$- and $z$-dimensions.

3. The wave-induced shift in wind direction can lead to considerable changes in the energy harvesting of individual wind turbines and the whole wind farm by wake deflection and yawing control. Therefore, this should not be neglected in future numerical studies and engineering models for offshore wind energy.

4. The absolute variations in energy production for individual wind turbines due to waves decrease progressively downstream. Notably, the relative change in total power output can be as significant as an increase of $20.0\%$ in the wind-following wave scenario and a decrease of $27.3\%$ in the wind-opposing wave scenario. These scenarios, characterized by moderate wind speeds and fast waves, are commonly observed in the North Sea area.

*Code and data availability.* The PALM INPUT files, OUTPUT files, and plot scripts are available at https://doi.org/10.5281/zenodo.10890846. The PALM code is available at https://gitlab.palm-model.org/releases/palm_model_system. The USER_CODE for the wave-induced stress parameterization is available on reasonable request.

*Author contributions.* Xu Ning: Writing – original draft, Writing – review & editing, Validation, Software, Methodology, Investigation, Formal analysis, Conceptualization. Mostafa Bakhoday-Paskyabi: Final review, Project administration, Funding acquisition.

*Competing interests.* The author has declared that there are no competing interests.

*Acknowledgements.* This work is part of the LES-WIND project and the authors would like to acknowledge the funding from the academic agreement between University of Bergen and Equinor. The simulations were performed on resources provided by UNINETT Sigma2 - the



National Infrastructure for High Performance Computing and Data Storage in Norway. We acknowledge that AI technology is utilized only for improving text readability, including sentence structure and word choice, to facilitate better understanding by readers.





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
