# Peer review of "Swell Impacts on an Offshore Wind Farm in Stable Boundary Layer: Wake Flow and Energy Budget Analysis"

_Wind Energy Science, 2024_

## Referee Comment (RC1)

**Report on manuscript:**
**Swell Impacts on an Offshore Wind Farm in Stable Boundary Layer: Wake Flow and Energy Budget Analysis**

Ning et al.

**Summary**

The paper quantifies the effect of swells (waves traveling faster than the local winds) using a wave-induced stress parameterization. The parameterization is implemented in an open-source Large Eddy Simulation (LES) code and used to study the impact of wave-induced stress in the wake flow and power output of a real offshore wind farm under a stable atmospheric boundary layer. Two cases are investigated, namely, wind following swell and wind opposing swell, and a detailed kinetic energy budget analysis is performed to quantify the direct and indirect wave-induced components. The paper is sufficiently detailed, and the discussions are a valuable addition to the community. In particular, the results showing the wind velocity profiles, wind direction, and budgets are interesting and the paper can be published in WES. I have a few major concerns regarding the wave model, and minor comments on adding some useful references to recent papers for wave-modeled LES simulations and ML methods for offshore wind farms.

**Major Comments**

- My major concern lies with the wave model. To parameterize the wave-induced stress, the authors use an empirical wave damping rate from Ardhuin et al. (2010), in conjunction with a given wave spectrum. Are there any validation studies, without wind farms, where such an approach is valid? There seem to be two tuning constants, 1) the parameter $f_e$, and the decay coefficient for the wave surface stress $a = 1$. From potential theory if $\tau_w \sim  \sim $, where the orbital velocity decays as $(u, w)_{orb} \sim \exp(-kz)$, shouldn't the most obvious choice of $a$ be 2? What is the thought process behind choosing $a = 1$?

- By limiting the integral to $\omega_c$ the higher frequency wave contributions are accounted for in a roughness length. Why not have a similar growth rate (instead of a damping one) to account for this? There are also recent models such as [1] that calculate the stress due to wind waves. Maybe, a Charnock model is sufficient for the current work, but this is an area to consider.

- The Charnock constant $\alpha$ is another free parameter chosen here. Is there any rationale behind this particular value? For instance, see *Liu et al (2012)*[2] where they discuss the different values used for the constant in different models.

- Is the wave model turned on at the same time as the cooling rate, or with the neutral flow?

- In line 160, $z_0 = 0.0002$. Previously it was mentioned that $z_0$ is calculated using the Charnock model. This seems inconsistent. Or is the above $z_0$ only for the pre-runs?

- Stable boundary layer simulations are generally quasi-steady. What is the averaging window for the simulations? Is the window chosen over a range where $u_*$ is a constant? Maybe a plot of $u_*$ as a function of time, with the averaging window highlighted will be useful.

- If I understand the wave stress correctly, it is independent of the flow characteristics, and the wave shape is fixed. However, with the introduction of wind turbines, $u_*$ decreases, and the wave effect should be more pronounced as the wave age increases.

- In Figure 7, can anything be said about the wake decay, i.e. does the presence of waves result in longer wakes? Is the velocity deficit formally defined somewhere in the text (is it normalized?)?

**Minor Comments**

- In the introduction, while discussing CFD papers for offshore wind farms, I think it is worth adding a reference to a recent review paper by *Deskos et. al 2021* [3]

- In line 45, the two Yangs in *Yang et al. (2014) and (2022b)* are different. The authors should add a citation to *Xiao S & Yang D. 2019*[4] which is relevant.

- Above line 45, the authors point out that the shortcomings of the wave-averaged (roughness length) approach can be addressed using the wave-phase resolved approach. However, there exist wave phase-aware models that lie between these two approaches [1, 5], and ML-based approaches [6, 7] that are relevant.

- Can the rationale behind multiplying the Donelan Spectrum with the exponential factor be explained?

**References**

[1] Aiyer, A. K., L. Deike, and M. E. Mueller, 2022: A Sea Surface–Based Drag Model for Large-Eddy Simulation of Wind–Wave Interaction. J. Atmos. Sci., 80, 49–62.

[2] Liu B., C. Guan, and L. Xie (2012), The wave state and sea spray related parameterization of wind stress applicable from low to extreme winds, J. Geophys. Res., 117, C00J22

[3] Deskos, G., J. C. Y. Lee, C. Draxl, and M. A. Sprague, 2021: Review of Wind–Wave Coupling Models for Large-Eddy Simulation of the Marine Atmospheric Boundary Layer. J. Atmos. Sci., 78, 3025–3045

[4] Xiao S, Yang D. Large-Eddy Simulation-Based Study of Effect of Swell-Induced Pitch Motion on Wake-Flow Statistics and Power Extraction of Offshore Wind Turbines. Energies. 2019; 12(7):1246.

[5] A. K. Aiyer, L. Deike, M. E. Mueller; A dynamic wall modeling approach for large eddy simulation of offshore wind farms in realistic oceanic conditions. J. Renewable Sustainable Energy 1 January 2024; 16 (1): 013305.

[6] Zexia Zhang, Xuanting Hao, Christian Santoni, Lian Shen, Fotis Sotiropoulos, Ali Khosronejad, Toward prediction of turbulent atmospheric flows over propagating oceanic waves via machine-learning augmented large-eddy simulation, Ocean Engineering, Volume 280, 2023, 114759, ISSN 0029-8018,

[7] Yousefi K, Hora GS, Yang H, Veron F, Giometto MG. A machine learning model for reconstructing skin-friction drag over ocean surface waves. Journal of Fluid Mechanics. 2024;983:A9. doi:10.1017/jfm.2024.81

---

## Referee Comment (RC2)

**Review comments on "Swell Impacts on an OffshoreWind Farm in Stable BoundaryLayer:Wake Flow and Energy Budget Analysis" (wes-2024-38)**

The study introduces a novel parameterization for modeling the effects of waves on marine atmospheric boundary layer flows. This parameterization models wave-induced stress based on an empirical wave directional spectrum, incorporating it into a wall model that augments equilibrium wall shear stress. Using this wave-stress wall model, the authors investigate the impact of waves on wind farm performance under stable atmospheric conditions. Through detailed kinetic energy budget analysis, they demonstrate that waves influence energy advection, primarily impacting the flow indirectly rather than through direct wave-induced work. Additionally, the paper examines wave-induced shifts in wind direction and changes in wind speed and turbulence characteristics, which affect wake deflection and turbine output, especially in typical North Sea conditions with moderate wind and fast waves. This research offers valuable insights into offshore wind energy dynamics and presents findings that can be applied to optimize real-world wind farm performance. I therefore recommend the manuscript for publication, pending the authors' responses to the following comments and questions:

1. In section 2.2, the damping rate for the wave-induced stress is based on whether the flow is laminar or turbulent which is distinguished using the critical Reynolds number.

   - Why is $2 \times 10^5/H_s$ the critical Reynolds number?
   - What is the value of orbital velocity $u_{orb}$ or equivalently the Reynolds number ($Re$) prescribed in the simulation? Is $u_{orb}$ prescribed as a constant value or does it vary horizontally such that it alters flow state between laminar and turbulent conditions?
   - $c_p$ in Eq. 6 is not defined anywhere in the manuscript.

2. Section 2.3 is incomplete. Additional discussion should talk about how is Eq. (11) implemented as a wall model in the LES code. How is the friction velocity $u_*$ determined from the wall model?

3. line 125: "The sum of viscous and turbulent stresses ......

$$\boldsymbol{\tau}_{tot} - \boldsymbol{\tau}_w = K_m \frac{\partial \boldsymbol{u}(z)}{\partial z}, z < 10 \ m, K_m = \kappa u_* z$$

"

This statement is incorrect because $\boldsymbol{\tau}_\nu = \nu \partial \boldsymbol{u}(z)/\partial z$ is the viscous stress and the eddy viscosity $K_m = \kappa u_* z$ only models the turbulent stress $\boldsymbol{\tau}_t = K_m \partial \boldsymbol{u}(z)/dz$. Eq. 10 should rather be written as

$$\boldsymbol{\tau}_{tot} - \boldsymbol{\tau}_w = (\nu + K_m) \frac{\partial \boldsymbol{u}(z)}{\partial z}, z < 10 \ m, K_m = \kappa u_* z$$

4. $\tau_{t,ij}$ in Eq. (2) refers to SGS stress while $\tau_t$ in Eq. (9) represents Reynolds shear stress. Using same notation for two different stress would confuse the readers. I recommend replacing $\tau_{t,ij}$ in Eq. (2) with $\tau_{SGS,ij}$.

5. Throughout the manuscript replace "swell waves" with just "swell/swells".

6. Comments on simulation abbreviations (M0-3):

   - I recommend authors to create a table and summarize the different simulation cases in the table. Readers can always come back and refer the table for what the notations M0,M1, M2, M3 .etc. mean.

   - Caption in figure 4(g) uses notation P0-3 while it should be M0-3.

   - I also recommend authors to use a different naming convention. M0-3 is difficult to follow through and can confuse readers. I suggest renaming M0 → WFW-pre (wind following wave prerun), M1 → WOW-pre (wind opposing wave prerun), M2 → WFW-C12 (wind following wave with 12 m/s peak phase speed), and M3 → WOW-C10 (wind opposing wave with 10 m/s peak phase speed).

7. Some additional comments regarding corrections to the text are directly annotated on the PDF of the manuscript and merged with this review report.

[revised manuscript text omitted]

---

## Author Comment (AC1)

**Response to Reviewers**

**Title:** Swell Impacts on an Offshore Wind Farm in Stable Boundary Layer: Wake Flow and Energy Budget Analysis

**Manuscript number:** WES-2024-38

We take this opportunity to thank the editor and reviewers of our paper for their kind collaboration to the improvement of this manuscript. We have taken into account all the concerns raised and we have made suggested modifications, marked by yellow background in the revised manuscript.

**Reviewer comments**

**Reviewer #1**

**General comments:** The paper quantifies the effect of swells (waves traveling faster than the local winds) using a wave-induced stress parameterization. The parameterization is implemented in an open-source Large Eddy Simulation (LES) code and used to study the impact of wave-induced stress in the wake ow and power output of a real o shore wind farm under a stable atmospheric boundary layer. Two cases are investigated, namely, wind following swell and wind opposing swell, and a detailed kinetic energy budget analysis is performed to quantify the direct and indirect wave-induced components. The paper is sufficiently detailed, and the discussions are a valuable addition to the community. In particular, the results showing the wind velocity pro les, wind direction, and budgets are interesting and the paper can be published in WES. I have a few major concerns regarding the wave model, and minor comments on adding some useful references to recent papers for wave-modeled LES simulations and ML methods for offshore wind farms.

**Response:** We greatly appreciate your recognition of our work and the constructive suggestions. We have carefully addressed each of your concern and have provided detailed responses below.

**Major Comments:**

1. My major concern lies with the wave model. To parameterize the wave-induced stress, the authors use an empirical wave damping rate from Ardhuin et al. (2010), in conjunction with a given wave spectrum. Are there any validation studies, without wind farms, where such an approach is valid? There seem to be two tuning constants, 1) the parameter $f_e$, and the decay coefficient for the wave surface stress $a = 1$. From potential theory if $\tau_w < u'w' >< u_{orb}w_{orb} >$, where the orbital velocity decays as $(u, w)_{orb} \, exp\,(-kz)$, shouldn't the most obvious choice of $a$ be 2? What is the thought process behind choosing $a = 1$?

**Response:** Thank you for your insightful questions and observations. We performed this validation in our previous work, "Parameterization of Wave-Induced Stress in Large-Eddy Simulations of the Marine Atmospheric Boundary Layer" (recently published) [1]. In that study, we conducted five groups of simulations (24 cases in total) using the same wave parameterization approach as in this study and compared the results against both wave-phase-resolved LES [2] and measurement data [3,4]. Our findings demonstrated that our model could reproduce typical swell-induced flow features, such as upward momentum fluxes and low-level jets, without explicitly resolving the wave surface geometry.

As you pointed out, the estimation of model parameters, particularly the wave damping rate $\beta^d$ and decay coefficient $a$, is crucial for accurate prediction. In our previous work, we leveraged wave-phase-resolved simulation results to calibrate $\beta^d$ and discussed four different methods for estimating the wave damping rate (see Fig. 20 in [1]). We found that while none of the methods perfectly accounts for all influencing factors, the empirical expression from Ardhuin et al. [12] (with $f_e$ calculated using Eq. 26 in [1]) generally provided reasonable wave damping rates close to our calibrated values in most wind-wave regimes, including the conditions set up in this study. Thus, based on our experience in previous work, we decided to use Ardhuin's method with $f_e = 0.008$ in the current study.

Regarding decay coefficient $a$, I agree with you that the potential theory suggests $a = 2$, while the study of Wu [7] shows that $a$ can vary between 0.5 and 4.0, influenced by multiple wind-wave factors and the decay rates of different wave-induced perturbated components, $\tilde{u}, \tilde{w}$, even differ from each other (their Fig. 11). Another estimation can be found from the wave-phase-resolved simulation results of Jiang [2], where the decay coefficient $a$ varies mostly below 2.0 and around 1.0 ($\alpha_p$ in their Table 2 and Table 3) for similar wind-wave conditions to those in our cases. Therefore, in this study, we use a value of 1.0 as a reasonable approximation, though we acknowledge that more observational data are needed to refine this parameter. We hope this clarifies our rationale and provides context for our parameter choices.

2. By limiting the integral to $\omega_c$ the higher frequency wave contributions are accounted for in a roughness length. Why not have a similar growth rate (instead of a damping one) to account for this? There are also recent models such as Aiyer et al. (2023) that calculate the stress due to wind waves. Maybe, a Charnock model is sufficient for the current work, but this is an area to consider.

**Response:** Thank you for the valuable suggestion. This is also considered in Section 2.3.2 in our previous work [1], where the stresses induced by wind waves, $\tau_{ww}$, were calculated by integrating over the portion of the wave spectrum with frequencies higher than $\omega_c$ (see Eq. 13 in [1]). Moreover, we examined how the decay coefficient for these stresses influences the wind profile and total momentum flux, comparing our results with empirical curves derived from the COARE measurement campaign [8] (see Fig. 3 and Fig. 4 in [1]). Our findings suggested that the most effective approach for handling $\tau_{ww}$ is to treat it as a friction force acting only at the surface, rather than as a vertical profile with an exponential decay. In other words, using a decay coefficient $a = \infty$ yields the best agreement with observations.

In this case, $\tau_{ww}$ does not explicitly contribute to the constant flux layer equation (Eq. 11 in the current study), and the total stress and friction velocity can be determined using the roughness length $z_0$ and the swell-induced stress (please also see our response to your 7th comment). Furthermore, given that we only considered the scenarios with low wind speed and strong waves in this study, the Charnock model serves as a sufficient approximation for representing the combined effects of wind wave stress and surface viscosity in the current study.

It should be noted that under strong wind conditions, the Charnock relationship may not sufficiently capture windsea-induced stress, as indicated by Liu et al. [13]. Integrating our proposed model with more advanced approaches for modeling wind-wave stress, such as those presented by Aiyer et al. [14], could offer a promising direction for future research. This is particularly relevant for scenarios involving complex wind-wave interactions.

3. The Charnock constant $\alpha$ is another free parameter chosen here. Is there any rationale behind this particular value? For instance, see Liu et al (2012) where they discuss the

different values used for the constant in different models.

**Response:** Thank you for bringing up this important point. Previous studies based on diverse observational datasets have demonstrated that the Charnock constant can indeed vary considerably with wind speed and wave age. In our study, we selected the Charnock constant based on the version 3.0 of the Coupled Ocean–Atmosphere Response Experiment (COARE) bulk algorithm [8]. The COARE 3.0 work provides a thorough summary of prior observational work on the Charnock parameter in Section 3.c and reveals a trend where the Charnock constant increases with wind speed above 10 m/s but scatters around 0.01 for lower wind speeds. They recommend a constant value of 0.011 for scenarios where $U_{10} < 10$ m/s. In our case, we choose a slightly higher value of 0.012, lying between their recommendation and other observational data. Furthermore, in another of our previous studies evaluating the performance of various roughness length parameterizations, we observed that variations in roughness length had a minimal impact on the flow field under low wind speed conditions (as shown in Fig. 4 and Fig. 5) [11]. Based on these findings, we believe that any potential errors associated with the chosen Charnock parameter value in the current study are likely negligible.

4. Is the wave model turned on at the same time as the cooling rate, or with the neutral flow?

**Response:** Thank you for your question. To prevent inertial oscillations caused by the introduction of swell forcing under stable conditions, the wave parameterization is not activated at the start of the simulation. Instead, it is turned on at $t = 45$ hours, which is 9 hours after surface cooling begins. The strength of the swell-induced stress is linearly ramped up, reaching the full value specified by Eq. 3 at $t = 47$ hours, and then lasts one more hour until the end of the preruns. In the main runs, both the surface cooling and wave stress are active from the beginning and continue throughout the simulation. We have rephrased the manuscript to clarify this setup.

**Corresponding revision:**

Page 8, first paragraph.

5. In line 160, $z_0 = 0.0002$. Previously it was mentioned that $z_0$ is calculated using the Charnock model. This seems inconsistent. Or is the above $z_0$ only for the pre-runs?

**Response:** Thank you for pointing out this inconsistency. In the wave parameterization, we use the Charnock model to calculate the roughness length dynamically. However,

in the control cases without wave parameterization, a constant roughness length of $z_0 = 0.0002$ is applied. This distinction has been clarified in Section 2.4 of the revised manuscript to avoid any confusion.

6. Stable boundary layer simulations are generally quasi-steady. What is the averaging window for the simulations? Is the window chosen over a range where $u_*$ is a constant? Maybe a plot of $u_*$ as a function of time, with the averaging window highlighted will be useful.

**Response:** We highly appreciate the reviewer's suggestion. To address your comment, we have included a figure depicting the time series of friction velocity to illustrate its temporal evolution as surface cooling and wave effects are introduced. The figure demonstrates that $u_*$ experiences an initial adjustment period of approximately 3 hours until the end of prerun (45h $<$ t $<$ 48h), during which it decreases/increases as the wind-following/wind opposing wave effects are gradually introduced (with linearly increasing strength). After this adjustment phase, $u_*$ stabilizes and remains nearly constant throughout the mainrun stage. We use a 1-hour averaging window for the final hour of each simulation, and the variations in $u_*$ within this window are $-0.3\%$, $-0.5\%$, and $-1.5\%$, for no wave, wind-following wave, and wind-opposing wave cases respectively. We hope this addition better illustrates that the mainrun cases have reached a quasi-steady state during the last hour.

[Figure]

7. If I understand the wave stress correctly, it is independent of the flow characteristics, and the wave shape is fixed. However, with the introduction of wind turbines, $u_*$ decreases, and the wave effect should be more pronounced as the wave age increases.

**Response:** Thank you for your insightful feedback. The wave shape is determined by the predefined empirical wave spectrum from Donelan, which remains fixed throughout each simulation. However, the influence of wind farms on the wave effects is dynamically accounted for in our model. The calculation of wave-induced stress and total stress requires both the wave field and the 10m wind speed as inputs. As described in Eq. 3, the integration range for calculating surface wave stress $\tau_w(0)$ is from 0 to $\omega_c$, where $\omega_c = g/U_{10}$ is a time-varying quantity.

Furthermore, as shown in Eq. 11, constant flux layer variables such as roughness length $z_0$, Monin-Obukhov length $L$, friction velocity $u_*$, and total stress $\tau_{tot}$ are recalculated iteratively at each surface grid point and at every timestep. This ensures that the effects of the wind farm, including reductions in $u_*$ and changes in surface stresses, are dynamically integrated into the parameterization. To clarify this process, we have added a detailed description and a flowchart (see Fig. 1 in the revised manuscript) that illustrates how the wave parameterization is incorporated into the wall-stress model and how these variables in the constant flux layer interact.

**Corresponding revision:**

Page 5, Fig. 1; Page 6, the last paragraph in Section 2.3.

8. In Figure 7, can anything be said about the wake decay, i.e. does the presence of waves result in longer wakes? Is the velocity deficit formally defined somewhere in the text (is it normalized?)?

**Response:** Thank you for your questions. The definition of velocity deficit is provided in the first paragraph on page 14. It is a normalized quantity calculated as $1.0 - u_h / u_{h,0}$, where $u_{h,0}$ represents the inflow wind speed at hub height. Unfortunately, due to the limited domain size along the x-axis, we are unable to capture the entire development of the wind farm wake flow. However, our results indicate that the presence of swell does not significantly impact the velocity deficit of the wake. Specifically, the wake width-averaged velocity deficit from cases with and without swell effects nearly overlap immediately behind the wind farm. Additionally, the turbulent kinetic energy (TKE) levels in the wake for all scenarios converge around x=20km, suggesting that the wake recovery further downstream is likely unaffected by the waves. The primary influence of swell is observed in the shift of the wake direction rather than in an extension/reduction of the wake length. We have updated the description for Fig. 9 to highlight this observation.

**Corresponding revision:**

Page 15, the last paragraph.

**Minor Comments:**

9. In the introduction, while discussing CFD papers for offshore wind farms, I think it is worth adding a reference to a recent review paper by Deskos et. al 2021.

**Response:** Thank you for this valuable suggestion. The review by Deskos et al. (2021) is indeed highly relevant to our topic and provides an excellent overview of wave-phase-averaged and wave-phase-resolved approaches, which are key to understanding the context and contributions of our work. We have incorporated this reference into the introduction to help readers better grasp these concepts and to situate our study within the broader field of CFD modeling for offshore wind farms. Thank you for bringing this important work to our attention.

**Corresponding revision:**

Page 2, the second paragraph.

10. In line 45, the two Yangs in Yang et al. (2014) and (2022b) are different. The authors should add a citation to Xiao S & Yang D. 2019 which is relevant.

**Response:** Thank you for bringing this to our attention. We have corrected the citations to ensure clarity regarding the two different Yangs. We have also included the relevant work of Xiao S. & Yang D. (2019) in the revised manuscript.

**Corresponding revision:**

Page 2, the second paragraph.

11. Above line 45, the authors point out that the shortcomings of the wave-averaged (roughness length) approach can be addressed using the wave-phase resolved approach. However, there exist wave phase-aware models that lie between these two approaches (Aiyer et al. 2022, Aiyer et al. 2024), and ML-based approaches (Zhang et al. 2023, Yousefi et al. 2024) that are relevant.

**Response:** Thank you for highlighting these references that provide alternative

approaches lying between wave-averaged and wave-phase-resolved methods. We agree that they are highly relevant, and we have included and discussed them in the revised introduction section.

**Corresponding revision:**

Page 2, the second paragraph.

12. Can the rationale behind multiplying the Donelan Spectrum with the exponential factor be explained?

**Response:** Thank you very much for raising this concern. The exponential factor is applied to adapt Donelan's spectrum, originally designed for a fully developed windsea field, to represent a swell-dominated spectrum. Specifically, by letting the coefficient $\omega_0^{-3} = -0.01$, the factor $exp[-0.01\omega^3]$ effectively damps the high-frequency wave components (mimicking the dissipation of windsea waves during long-distance travel) while preserving the low-frequency swell., as shown in the following figure. This method was first introduced by Hanley et al. [9] (their Eq. 24) and later adopted by Chen et al. [10] (Eq. 13), although, as far as we know, there is no theoretical or experimental basis to justify the exact value of -0.01. It is important to note that the influence of this exponential factor on our study is minimal since the high-frequency part of the wave spectrum is approximated using the Charnock relation rather than direct integration.

[Figure]

**Reviewer #2**

**General comments:** The study introduces a novel parameterization for modeling the effects of waves on marine atmospheric boundary layer flows. This parameterization models wave-induced stress based on an empirical wave directional spectrum, incorporating it into a wall model that augments equilibrium wall shear stress. Using this wave-stress wall model, the authors investigate the impact of waves on wind farm performance under stable atmospheric conditions. Through detailed kinetic energy budget analysis, they demonstrate that waves in fluence energy advection, primarily impacting the flow indirectly rather than through direct wave-induced work. Additionally, the paper examines wave-induced shifts in wind direction and changes in wind speed and turbulence characteristics, which affect wake deflection and turbine output, especially in typical North Sea conditions with moderate wind and fast waves. This research offers valuable insights into offshore wind energy dynamics and presents findings that can be applied to optimize real-world wind farm performance. I therefore recommend the manuscript for publication, pending the authors' responses to the following comments and questions:

**Response:** Thank you sincerely for your positive feedback and thoughtful comments. Please refer to our responses to the specific comments below for a comprehensive discussion of the improvements and clarifications made in the manuscript.

1. In section 2.2, the damping rate for the wave-induced stress is based on whether the flow is laminar or turbulent which is distinguished using the critical Reynolds number.

- Why is $2 \times 10^5 / H_s$ the critical Reynolds number?
- What is the value of orbital velocity $u_{orb}$ or equivalently the Reynolds number (Re) prescribed in the simulation? Is $u_{orb}$ prescribed as a constant value or does it vary horizontally such that it alters flow state between laminar and turbulent conditions?
- $c_p$ in Eq. 6 is not defined anywhere in the manuscript.

**Response:** Thank you for your questions.

Regarding the critical Reynolds number, we adopted Ardhuin's approach (Section 2.b in [12]), which uses the expression $2 \times 10^5 / H_s$ as the threshold for distinguishing between laminar and turbulent flow states. According to Ardhuin et al., this criterion produced reasonable results, although they did not provide a detailed explanation or theoretical basis for this expression. We acknowledge that this is an important aspect that warrants further investigation through experiments and observations to establish a

more robust understanding, but addressing this goes beyond the scope of the present study.

In our implementation, the wave spectrum is prescribed and remains constant and homogeneous during the simulations. The significant surface orbital velocity $u_{orb}$ is calculated as

$$u_{orb} = \sqrt{m_2/(2m_0)}H_s, \quad H_s = 4\sqrt{m_0}$$

where $m_0$ and $m_2$ are the zeroth and second moments of the wave spectrum respectively. While the wave spectrum is held constant, the integration limits over the wave spectrum are $0 < \omega < \omega_c$, and $\omega_c$ varies with the 10 m wind speed. This means that $u_{orb}$ may also has temporal and spatial variations. In our wave-following and wave-opposing cases, the Reynolds numbers calculated using the averaged flow quantities from preruns are $2.0684 \times 10^5$ and $3.4755 \times 10^5$, respectively. Both are larger than the corresponding critical Reynolds numbers, indicating that the near-surface flow should predominantly remain turbulent condition throughout the entire domain.

We have added the definition of $c_p$ immediately after its first mention in the manuscript for clarity.

**Corresponding revision:**

Page 4, the second line after Eq. 4, the first line after Eq. 6.

2. Section 2.3 is incomplete. Additional discussion should talk about how is Eq. (11) implemented as a wall model in the LES code. How is the friction velocity $u_*$ determined from the wall model?

**Response:** Thank you for your comment. In Eq. 11, the total surface stress $\tau_{tot}$ and consequently the friction velocity $u_*$, have to be iteratively solved at $z = 10$m for each grid point, since the roughness length $z_0$ and Monin-Obukhov length also depend on $u_*$. This calculation is based on the known 10 m wind speed and the wave stress derived using Eq. 3. A more detailed description and a flowchart (Fig. 1 in the revision) have been added in Section 2.3. We believe this addition will help clarify the implementation of the proposed wave parameterization and how it is incorporated with the wall-stress model in the LES code.

**Corresponding revision:**

Page 5, Fig. 1; Page 6, the last paragraph in Section 2.3.

3. The line 125: "The sum of viscous and turbulent stresses ......

$$\tau_{tot} - \tau_w = K_m \frac{\partial u(z)}{\partial z}, z < 10m, K_m = \kappa u_* z$$

"

This statement is incorrect because $\tau_v = v \partial u(z) / \partial z$ is the viscous stress and the eddy viscosity $K_m = \kappa u_* z$ only models the turbulent stress $\tau_t = K_m \partial u(z) / \partial z$. Eq. 10 should rather be written as

$$\tau_{tot} - \tau_w = (v + K_m) \frac{\partial u(z)}{\partial z}, z < 10m, K_m = \kappa u_* z$$

**Response:** Thank your for pointing out the imprecise expression in our original statement. We have changed Eq. 10 into

$$\tau_{tot} - \tau_v - \tau_w = K_m \frac{\partial u(z)}{\partial z}, z < 10m, K_m = \kappa u_* z.$$

Here the viscous stress cannot be directly resolved and is assumed neglectable because of the high Reynolds number flow in our scenarios. This is a common simplification used in previous wave parameterization studies, such as in [3, 5, 10]. Thank you for identifying this issue and for your correction.

**Corresponding revision:**

Page 5, Eq. 10.

4. $\tau_{t,ij}$ in Eq. (2) refers to SGS stress while $\tau_t$ in Eq. (9) represents Reynolds shear stress. Using same notation for two different stress would confuse the readers. I recommend replacing $\tau_{t,ij}$ in Eq. (2) with $\tau_{SGS,ij}$.

**Response:** Sincere thanks for pointing out the inconsistency in our notation. We have now updated the notation in Eq. (2), replacing $\tau_{t,ij}$ with $\tau_{SGS,ij}$ to clearly differentiate between the SGS stress and the Reynolds shear stress.

**Corresponding revision:**

Page 3, Eq. 2.

5. Throughout the manuscript replace "swell waves" with just "swell/swells".

**Response:** Thank you for the suggestion. We have revised the manuscript accordingly, replacing "swell waves" with "swell" or "swells".

**Corresponding revision:**

Page 2, the first paragraph; Page 3, the second paragraph; Page 4, the last paragraph; Page 8, the first paragraph; Page 9, the first paragraph; Page 23, the third paragraph; Page 24, the last paragraph.

6. Comments on simulation abbreviations (M0-3):
- I recommend authors to create a table and summarize the different simulation cases in the table. Readers can always come back and refer the table for what the notations M0,M1, M2, M3 .etc. mean.
- Caption in figure 4(g) uses notation P0-3 while it should be M0-3.
- I also recommend authors to use a different naming convention. M0-3 is difficult to follow through and can confuse readers. I suggest renaming M0 → WFW-pre (wind following wave prerun), M1 →WOW-pre (wind opposing wave prerun), M2 → WFW-C12 (wind following wave with 12 m/s peak phase speed), and M3 → WOW-C10 (wind opposing wave with 10 m/s peak phase speed).

**Response:** Thank you for your valuable suggestions to improve the clarity of the simulation case abbreviations. We have added a table to the manuscript that lists abbreviations and wind-wave parameters for each case, providing a quick reference for readers. Following your recommendation, we have revised the naming scheme for the mainruns: M0, M1, M2, and M3 are now renamed as WFW-CTRL, WOW-CTRL, WFW-C12, and WOW-C10, respectively. Additionally, the caption for preruns in Figure 4(g) has been corrected to CTRL, WFW, and WOW. These changes have been implemented throughout the manuscript to enhance clarity and reduce potential confusion.

**Corresponding revision:**

Page 9, Table 1; Page 12, Fig. 6.

7. Some additional comments regarding corrections to the text are directly annotated on the PDF of the manuscript and merged with this review report.

**Response:** Thank you for your detailed annotations and corrections. We have reviewed

all the comments and revised the manuscript accordingly.

**Corresponding revision:**

Abstract; Page 4, the second paragraph; Page 6, the last paragraph; Page 17, the first paragraph; Page 18, caption of Fig. 12;

**References:**

1. Ning, X., & Paskyabi, M. B. (2024). Parameterization of wave-induced stress in large-eddy simulations of the marine atmospheric boundary layer. *Journal of Geophysical Research: Oceans*, *129*(9), e2023JC020722.
2. Jiang, Q., Sullivan, P., Wang, S., Doyle, J., & Vincent, L. (2016). Impact of swell on air–sea momentum flux and marine boundary layer under low-wind conditions. *Journal of the Atmospheric Sciences*, *73*(7), 2683-2697.
3. Semedo, A., Saetra, Ø., Rutgersson, A., Kahma, K. K., & Pettersson, H. (2009). Wave-induced wind in the marine boundary layer. *Journal of the Atmospheric Sciences*, *66*(8), 2256-2271.
4. Smedman, A., Högström, U., Sahlée, E., Drennan, W. M., Kahma, K. K., Pettersson, H., & Zhang, F. (2009). Observational study of marine atmospheric boundary layer characteristics during swell. *Journal of the atmospheric sciences*, *66*(9), 2747-2763.
5. Song, J., Fan, W., Li, S., & Zhou, M. (2015). Impact of surface waves on the steady near-surface wind profiles over the ocean. *Boundary-Layer Meteorology*, *155*, 111-127.
6. Chen, S., Xue, Y., Yang, B., Yu, Y., & Qiao, F. (2023). Observation and analysis of the influence of wind waves on air-sea momentum fluxes. *Science China Earth Sciences*, *66*(7), 1547-1555.
7. Wu, L., Hristov, T., & Rutgersson, A. (2018). Vertical profiles of wave-coherent momentum flux and velocity variances in the marine atmospheric boundary layer. *Journal of Physical Oceanography*, *48*(3), 625-641.
8. Fairall, C. W., Bradley, E. F., Hare, J. E., Grachev, A. A., & Edson, J. B. (2003). Bulk parameterization of air–sea fluxes: Updates and verification for the COARE algorithm. *Journal of climate*, *16*(4), 571-591.
9. Hanley, K. E., & Belcher, S. E. (2008). Wave-driven wind jets in the marine atmospheric boundary layer. *Journal of the Atmospheric Sciences*, *65*(8), 2646-2660.
10. Chen, S., Qiao, F., Jiang, W., Guo, J., & Dai, D. (2019). Impact of surface waves on wind stress under low to moderate wind conditions. *Journal of Physical Oceanography*, *49*(8), 2017-2028.

11. Ning, X., Paskyabi, M. B., Bui, H. H., & Penchah, M. M. (2023). Evaluation of sea surface roughness parameterization in meso-to-micro scale simulation of the offshore wind field. *Journal of Wind Engineering and Industrial Aerodynamics*, *242*, 105592.
12. Ardhuin, F., Rogers, E., Babanin, A. V., Filipot, J. F., Magne, R., Roland, A., ... & Collard, F. (2010). Semiempirical dissipation source functions for ocean waves. Part I: Definition, calibration, and validation. Journal of Physical Oceanography, 40(9), 1917-1941.
13. Liu, Bin, Changlong Guan, and Lian Xie. "The wave state and sea spray related parameterization of wind stress applicable from low to extreme winds." *Journal of Geophysical Research: Oceans* 117.C11 (2012).
14. Aiyer, Aditya K., Luc Deike, and Michael E. Mueller. "A sea surface–based drag model for large-eddy simulation of wind–wave interaction." *Journal of the Atmospheric Sciences* 80.1 (2023): 49-62.